# Importance of SOA formation of α-pinene, limonene and *m*-cresol comparing day- and nighttime radical chemistry

Anke Mutzel[1,a], Yanli Zhang[1,2], Olaf Böge[1], Maria Rodigast[1,a], Agata Kolodziejczyk[3,1], Xinming Wang[2] and Hartmut Herrmann[1]

[1]Leibniz Institute for Tropospheric Research (TROPOS), Atmospheric Chemistry Department (ACD), Permoserstr. 15, 04318 Leipzig, Germany

[2]State Key Laboratory of Organic Geochemistry and Guangdong Key Laboratory of Environmental Protection and Resources Utilization, Guangzhou Institute of Geochemistry, Chinese Academy of Sciences, Guangzhou 510640, China

[3]Institute of Physical Chemistry of the Polish Academy of Sciences, Kasprzaka 44/52, 01-224 Warsaw, Poland

[a] now at Eurofins Institute Dr. Appelt Leipzig, Täubchenweg 28, 04318 Leipzig

[b] now at: Indulor Chemie GmbH & Co. KG Produktionsgesellschaft Bitterfeld, 06749 Bitterfeld-Wolfen, Germany

*Correspondence to*: Anke Mutzel (mutzel@tropos.de ) and Hartmut Herrmann (herrmann@tropos.de )

## Abstract

The oxidation of biogenic and anthropogenic compounds leads to the formation of secondary organic aerosol mass (SOA). The present study aims to investigate α-pinene, limonene and *m*-cresol with regards to their SOA formation potential dependent on relative humidity (RH) under night- (NO$_3$ radicals) and daytime conditions (OH radicals) and the resulting chemical composition. It was found that SOA formation potential of limonene with NO$_3$ under dry conditions significantly exceeds the one of the OH radical reaction, with SOA yields of 15 – 30 % and 10 – 21 %, respectively. Additionally, the nocturnal SOA yield was found to be very sensitive towards RH, yielding more SOA under dry conditions. On the contrary, the SOA formation potential of α-pinene with NO$_3$ slightly exceeds that of the OH radical reaction, independent from RH. In average, α-pinene yielded SOA with about 6 - 7% from NO$_3$ radicals and 3 – 4 % from OH radical reaction. Surprisingly, unexpected high SOA yields were found for *m*-cresol oxidation with OH radicals (3 - 9%) with the highest yield under elevated RH (9%) which is most likely attributed to a higher fraction of 3-methyl-6-nitro-catechol (MNC). While α-pinene and *m*-cresol SOA was found to be mainly composed of water-soluble compounds, 50 – 68 % of nocturnal SOA and 22 - 39% of daytime limonene SOA is water-insoluble. The fraction of SOA-bound peroxides which originated from α-pinene varied between 2 – 80% as a function of RH.

Furthermore, SOA from α-pinene revealed pinonic acid as the most important particle-phase constituent under day- and nighttime conditions with fraction of 1 – 4%. Other compounds detected are norpinonic acid (0.05 – 1.1% mass fraction), terpenylic acid (0.1 – 1.1 % mass fraction), pinic acid (0.1 – 1.8 % mass fraction) and 3-methyl-1,2,3-tricarboxylic acid (0.05 – 0.5 % mass fraction). All marker compounds showed higher fractions under dry conditions when formed during daytime and showed almost no RH effect when formed during night.

# 1 Introduction

Large amounts of volatile organic compounds (VOC) are emitted into the atmosphere from both biogenic and anthropogenic sources with estimated source strengths of about 1300 TgC yr$^{-1}$ (Goldstein and Galbally, 2007). Once emitted, VOC undergo gas-phase reactions with ozone ($O_3$), hydroxyl (OH) or nitrate ($NO_3$) radicals (Atkinson and Arey 2003). Those reactions result in the formation of oxygenated products, with a lower vapor pressure than the parent hydrocarbons, which are subject to partitioning into the particle phase leading to the formation of secondary organic aerosol (SOA). The atmospheric degradation of biogenic volatile organic compounds (BVOCs) and subsequent SOA formation was subject of numerous studies during the last decades (Hallquist et al., 2009, Glasius and Goldstein 2016, Shrivastava et al., 2017). The majority of these studies examined the reaction initiated by the OH radical or ozone as they are considered as most dominating VOC sinks, although measurements indicated $NO_3$ radical reaction is the most important sink for several VOCs during nighttime (Geyer et al., 2001). It was demonstrated that $NO_3$-radical initiated oxidation contributes with 28% to the overall VOC conversion compared to 55 % for OH radical reaction and 17% for the ozonolysis (Geyer et al., 2001, Kurtenbach et al., 2002, McLaren et al., 2010, Liebmann et al., 2018, Liebmann et al., 2018). While $NO_2$ and $O_3$ serve as precursor for nitrate radicals, $NO_3$ is most dominating at night due to the fast photolysis and degradation with NO (Wayne et al., 1991, Brown and Stutz, 2012). The number of studies interconnecting $NO_x$ and BVOC emissions (Fry et al., 2009, Xu et al., 2015) are increasing, because the reaction with $NO_3$ is often considered to be more important for BVOCs than for AVOCs (Brown and Stutz, 2012).

Even though the number of studies, investigating the rise of $NO_3$ radical-initiated SOA formation, during the last few years increased (e.g., Pye et al., 2010, Fry et al., 2014, Boyd et al., 2015, Fry et al., 2018, Qin et al., 2018, Joo et al., 2019), there is still an enormous lack of data with respect to SOA yields, the influence of RH on SOA formation and the product distribution in the gas and particle phase. Kinetic studies have shown rate constants for $\alpha$-pinene and limonene with $NO_3$ in the range of 1.1 – 6.5 x 10$^{-12}$ and 1.1 – 94 x 10$^{-11}$ cm$^3$ molecule$^{-1}$ s$^{-1}$ (Atkinson et al., 1984, Dlugokencky and Howard 1989, Barnes et al., 1990, Kind et al., 1998, Martinez et al., 1998, Martinez et al., 1999, Stewart et al., 2013). For $m$-cresol only two rate constants are reported in the range of 7.0 – 9.2 x 10$^{-12}$ cm$^3$ molecule$^{-1}$ s$^{-1}$ (Carter et al., 1981, Atkinson et al., 1984). Accordingly, at least for nighttime and on a regional scale, $NO_3$ reaction might lead to important contributions to VOC degradation and SOA formation. According to the comprehensive review, by Ng et al., 2017, $NO_3$ + VOC is worth investigating because: i) it can lead to anthropogenically influenced biogenic secondary organic aerosol (BSOA, Hoyle et al., 2007), ii) SOA yields might be higher than from OH and ozone (Ng et al., 2017), iii) it compromises an important source for organonitrates that serve as $NO_x$ and $NO_y$ reservoirs (von Kuhlmann et al., 2004, Horowitz et al., 2007) and iv) in a few regions it was identified as the most dominating SOA contributor (Hoyle et al., 2007, Pye et al., 2010, Chung et al., 2012, Kiendler-Scharr et al., 2016).

This study is aimed to investigate three selected precursor compounds, namely $\alpha$-pinene and limonene as biogenic VOCs and $m$-cresol as aromatic VOC with regards to their SOA formation potential under nighttime ($NO_3$ radicals) and daytime conditions (OH radicals). While $\alpha$-pinene and limonene are important BVOCs, $m$-cresol is often related to biomass burning. The chemical composition of formed SOA was characterized for their fraction of organic material (OM), water-soluble organic material (WSOM), SOA-bound peroxides and SOA marker compounds. For quantification of marker compounds, well known BSOA marker compounds (pinic acid,

pinonic acid etc.) were used while SOA originated from m-cresol was characterized using a SOA mix than contains mostly anthropogenic SOA compounds that are often related to biomass burning (Hoffmann et al., 2007). Furthermore, SOA yield and SOA growth will be discussed in detail as well as the influence of the relative humidity. The chemical composition of formed SOA was characterized for their fraction of organic material (OM), water-soluble organic material (WSOM), SOA-bound peroxides and SOA marker compounds.

## 2 Experimental

### 2.1 Chamber experiments

Experiments were conducted in the aerosol chamber under batch mode conditions at the Atmospheric Chemistry Department (ACD) of the Leibniz Institute for Tropospheric Research (TROPOS) in Leipzig. A brief description of the chamber will be given here because a complete description of the chamber can be found elsewhere

(Mutzel et al., 2016). The aerosol chamber is made of PTFE and is of cylindrical geometry with a total volume of 19 $m^3$ and a surface to volume ratio of 2 $m^{-1}$. The chamber is equipped with a humidifier to enable reactions at elevated RH and a temperature-controlled housing to keep the temperature stable at T = 298 K throughout the experimental run. The humidifier is connected to the inlet air stream to enable humidification of air entering the chamber. Experiments were conducted using ammonium sulfate/sulfuric acid seed (($NH_4$)$_2SO_4$/$H_2SO_4$) particles

of pH = 4 at RH 50 %). The seed particles were injected via a nebulizer without a dryer. Their RH-dependent pH-value was calculated by E-AIM (Clegg et al., 1998). All experiments were done with an initial hydrocarbon mixing ratio of 60 ppbv.

OH-radical reactions were initialized by photolysis of hydrogen peroxide ($H_2O_2$) in the presence of NO (10 ppb). $H_2O_2$ was continuously injected into the chamber with a peristaltic pump at 100 µL $hr^{-1}$ and was photolyzed with

UV-A-lamps (Osram Eversun Super). Applying the method developed by Barmet et al., 2011, the average OH radical mixing ratio in the chamber is about 3 – 5 x $10^6$ molecules $cm^{-3}$.

NO$_3$ radicals were produced in a pre-reactor (operated as flow tube) by the reaction of NO$_2$ and O$_3$. A fraction of the air flow (10 L $min^{-1}$) out of the total air flow in the flow tube (30 L $min^{-1}$) was directed to the chamber (Iinuma et al., 2010). Including the kinetic box model developed by Fry et al., 2014 into the COPASI

(COMPLEX PATHWAY SIMULATOR), the mixing ratio of NO$_3$ radicals is calculated for the present study to be 7.5 x $10^7$ molecules $cm^{-3}$. The reaction mechanism provided by Fry and co-workers we implemented into COPASI and the model was utilized to the aerosol chamber."

After a reaction time of 90 min the reaction was stopped and samples were taken passing chamber air over a 47 mm PTFE filter (borosilicate glass fiber filter coated with fluorocarbon, 47 mm in diameter, PALLFLEX

T60A20, PALL, NY, US) and QF filter (Micro-quartz fibre filter, 47 mm in diameter, MK 360, Munktell, Bärenstein, Germany), for 30 minutes at 30 L $min^{-1}$. During sampling time no additional air stream was added to chamber to avoid dilution. PTFE filters were quantified afterwards for biogenic and anthropogenic SOA marker compounds and QF to determine organic/elemental carbon (OC/EC), non-purgeable organic carbon (NPOC, formerly known as: water-soluble organic carbon) and for select experiments also concentration of inorganic

nitrate (NO$_3^-$).

Experiments were conducted either under nighttime conditions with NO$_3$ radicals or with OH radicals to represent daytime chemistry. A complete overview of all experiments can be found in Table 1.

Dilution rates and wall losses were considered as follows: NO$_3$ radicals and $H_2O_2$ were injected into the chamber with a bypass air of 10 L $min^{-1}$ and 5 L $min^{-1}$. Based on a reaction time of 90 minutes a dilution of 4.7 % (NO$_3$)

and 2.4 % (OH) can be estimated. These values are within the measurement uncertainty of the Proton-Transfer-Reaction Time-of-flight Mass Spectrometer (PTR-TOFM). According to the study by Romano and Hanna, 2018 an uncertainty of ±10% can be assumed and were thus not considered. Wall losses of VOCs were determined to be 2.5 x $10^{-5}$ $s^{-1}$ (α-pinene), 7.9 x $10^{-5}$ $s^{-1}$ (limonene) and 2.2 x $10^{-5}$ $s^{-1}$ (*m*-cresol). The consumption recorded by PTR-MS is corrected for those additional sinks. Particle wall losses were determined from the blank experiments

at RH = 50%. Time-dependent particle losses were used to correct the SMPS measurements. For blank experiments all compounds were injected into the chamber, except the hydrocarbon. Notably, wall losses at RH = 50% was used as approximation also for 0% and 75% RH, although losses might change under those condition according to their phase state. According to previous studies wall loss might be small due to the short reaction time. According to McMurry and Grosjean, a 90 minutes reaction time would result in a 10% loss of particles,

which is within the measurement uncertainty of the SMPS (McMurry and Grosjean, 1985). Additionally, seed particles were injected without a dryer. Consequently, they can be regarded as wet particles when they enter the chamber. Thus at RH =75% no additional loss is expected.

As the chamber is allowed to equilibrate for at least 10 minutes after seed injection, a dramatic wall loss under dry conditions would be directly observable in SMPS by a drastic decrease of particle volume with a constant

particle number. As this was not observed it can be assumed that wall loss at RH = 0% is in the same manner as at 50 %.

An ozone monitor was connected for all experiments. Specific conditions in the pre-reactor were set to avoid ozone entering the chamber during $NO_3$ radical reaction. Thereby for this reaction type ozonolysis as side reaction can be excluded. During $H_2O_2$ photolysis small amounts of $O_3$ is always formed, which might lead to

ozonolysis. It should be noted that OH radical reaction was conducted in the presence of $NO_x$. Thus formed $O_3$ will rapidly react with NO rather than with α-pinene and limonene. Due to low reaction rate constant and low concentration, ozonolysis occurs to a very small extent and cannot be excluded. A maximum $O_3$ concentration of 5 ppb was observed.

## 2.2 Online instrumentation

The consumption of precursor compounds (ΔHC) was monitored by a proton-transfer-reaction time-of-flight mass spectrometer (PTR-TOFMS, Ionicon, Lindinger et al., 1998). The particle size distribution was measured by a scanning mobility particle sizer (SMPS, Wiedensohler et al., 2012). In absence of reliable density estimation, an average density of 1 g $cm^{-3}$ was used to convert the SMPS measurement data into the increase in organic mass (ΔM). In absence of reliable density estimation, an average density of 1 g $cm^{-3}$ was used to convert

the SMPS measurement data into the increase in organic mass (ΔM). This assumption was also made for OM and NPOM measurement. The assumed density was not changed with RH. As the ft between OM and ΔM stays almost constant, the RH seems not to affect the density in the conducted experiments. The particle growth by water uptake was taken into account by collecting particles on filter and determine the content of organic material (OM). For most of the experiments it was found that both values (ΔM and OM) fit well indicating that

particle growth is mainly caused by organics rather than water. Monitors for ozone (49c ozone analyzer, Thermo Scientific, USA) and $NO_x$ (42i TL, Trace level NOx analyzer. Thermo Scientific, USA) were connected to the chamber as well.

### 2.3 Offline measurements

### 2.3.1 OC/EC, NPOC, inorganic nitrate and SOA bound peroxides

The quartz filter was cut into halves. One half was used for OC/EC quantification and the second was used for water-soluble organic carbon. The content of OC/EC was determined with a C-mat 5500 carbon analyzer applying a two-step thermographic method (Neusüß et al., 2002). The fraction of water-soluble organic carbon was determined as non-purgeable organic carbon with a TOC-V$_{CPH}$ analyzer (van Pinxteren et al., 2009). To do so, the second half of the QF was extracted in 25 mL ultrapure water for 30 minutes with an orbital shaker. The

resulting extract was filtered through a 0.45 µm syringe filter (Acrodisc 13, Pall, USA). 250 µL of the extract was used for $NO_3^-$ analysis. After acidification and sparging with $N_2$, the remaining extract was injected into the TOC-analyzer. The amount of $NO_3^-$ was determined by ion chromatography coupled with conductivity detection (IC-CD) using an AS18 column combined with AG18 guard column.

    For SOA bound peroxides, half of the PTFE filter was used. One quarter of the filter was used for the peroxide

test and the second quarter filter to determine the blank value. The method is described in detail elsewhere (Mutzel et al., 2013).

### 2.3.2 Sample preparation for LC/MS

    The sample preparation follows the method described in the literature (Hoffmann et al., 2007, Mutzel et al., 2015). Briefly, half of the PTFE filter was cut into small pieces and transferred into an extraction vial. 500 µL of

methanol was added and the vial was placed in an orbital shaker for 15 min at 1000 rotations min$^{-1}$. Insoluble material was removed by a syringe filter (0.2 mm, Acrodisc Pall, USA). Afterwards, the extraction was repeated with 500 mL of MeOH. The combined extracts were dried under a gentle stream of nitrogen and reconstituted in 250 mL of $CH_3OH/H_2O$ (50/50, v/v).

### 2.3.3 Analysis with HPLC/(-)ESI-TOFMS

A high-performance liquid chromatography (HPLC, Agilent, 1100 Series, Santa Clara, CA, USA) connected to a electrospray ionization time of flight mass spectrometer (microTOF, Bruker Daltonics, Bremen, Germany) was used for separation and quantification of marker compounds. For the separation an Agilent ZORBAX C18 column (3.0 x 150 mm, 5 µm particle size) was used at a temperature of 25°C and a flow rate of 0.5 ml min$^{-1}$ with 0.1 % acetic acid in ultrapure water (A) and 100 % methanol (B) as eluents. The gradient was as follows: 10

% B for 2 minutes, increased from 10 % B to 100 % B in 20 minutes, then held constant for 3 minutes and re-equilibrated for 5 minutes back to the initial conditions. The quantification was done in the negative ionization mode with a mass range between $m/z$ 50 and 1000 applying a series of sodium acetate clusters to calibrate mass accuracy. Quantification was done using authentic standard solutions within a 7-point calibration with three repetitions of each calibration point.

For anthropogenic SOA compounds, the separation was done as described above at 15°C and with 0.2 % acetic acid in water.

    The yield of the single compounds were calculated by taking the quantified amount from the filter, correcting for sampling volume. The numbers are given as fraction in formed organic mass

### 2.3.4 Chemicals

The following chemicals were used as received: α-pinene, limonene and *m*-cresol (Sigma-Aldrich, St. Louis, USA, purity 99%, 97% and 99%), terebic acid (Sigma-Aldrich, St. Louis, USA, purity 99 %) and pinic acid (Sigma-Aldrich, St. Louis, USA, purity 99%).

The following compounds were synthesized according to procedures given in the literature: norpinonic acid, terpenylic acid (Claeys et al., 2009), 3-methyl-1,2,3-butanetricarboxylic acid (Szmigielski et al., 2007),

diaterpenylic acid acetate (DTAA; Iinuma et al., 2009). The composition of the anthropogenic SOA mix is described in detail in Hoffmann et al., 2007.

## 3 Results and Discussion

### 3.1 SOA formation and yield

The SOA formation from the reaction of α-pinene, limonene and *m*-cresol with $NO_3$ radicals has been

investigated within this study with emphasis on SOA yields, the chemical composition in the particle phase, the influence of the RH and a final comparison to daytime chemistry with OH radicals. The SOA yields were calculated according to Odum et al., by calculating the amount or produced organic mass in relation to the amount of reacted hydrocarbon according to:

$$Y_{SOA} = \frac{\Delta M}{\Delta HC} \qquad \text{(Eq. 1)}$$


where

$\Delta M$ is the produced organic mass [µg m$^{-3}$]

$\Delta HC$ is the reacted amount of hydrocarbon [µg m$^{-3}$]

A complete overview about all experiments, the obtained results as well as the comparison to literature values, is given in Table 1. In general, the discussion is mainly focused on the amount of SOA mass produced after 90 minutes reaction time. Only the differences in curve shape of growth curve are discussed in detail in the respective section.

In general, α-pinene yielded higher SOA with $NO_3$ radicals ($Y_{NO3} \approx 6\%$) than with OH ($Y_{OH} \approx 3.5$ %). In the

case of limonene the difference is not as well discerned but can still be observed ($Y_{NO3} \approx 15 - 30$ %; $Y_{OH} \approx 10 - 21$ %). On the contrary, *m*-cresol yielded a dismissable amount of SOA with $NO_3$ radicals and moderate amounts with OH radicals ($Y_{OH} \approx 3 - 9$ %). Therefore, the highest SOA formation potential for $NO_3$ radical reaction was found for limonene and the lowest for *m*-cresol.

The SOA yield curves were parameterized according to Odum et al., 1996 following:


$$Y = \sum Y_i = M_0 \sum \frac{\alpha K_{OM,i}}{1 + K_{OM,i} M_0} \qquad \text{(Eq. 2)}$$

where

α is the mass yield of compound i

$K_{OM,i}$ is the partitioning coefficient of compound i

$M_0$ is the absorbing organic mass


By applying the one-product model approach, the fit produced very good results with $R^2 > 0.99$. The applicability of one-product models was also demonstrated by Friedmann and Farmer 2018. Yield curves without any effect of RH result in comparable $\alpha$ and K values. Those yield curves, with a distinct RH influence, show a higher partitioning coefficient for higher SOA yields together with increasing mass yields. All $\alpha$ and K values are depicted in the respective yield curves (Figure 1).

Only a limited number of studies provided parameterization of yield curves for VOC/OH/NOx and VOC/NO$_3$ according to Odum et al., 1996, which highlights the need for the present data set. Spittler et al., 2006 reported based on a two-product model for limonene/NO$_3$ $\alpha_{1/2}$ and $K_{1/2}$ values of 0.1249/0.3128 and 0.0348/0.0181. The reported values for $\alpha$ correspond well to values obtained in this study whereas K values are by one order of magnitude smaller. This variation could be caused by the different seed particles used, because Spittler and co-workers employed a pure organic seed and this study an inorganic seed was utilized.

Iinuma et al., 2010 reported based on a two-product model for cresol/OH $\alpha_{1/2}$ and K values of 0.1231/0.0004 and 0.0753. These values are in no agreement to reported values, which might be caused by different OH sources used.

*a-Pinene*

SOA yields for $\alpha$-pinene with NO$_3$ radicals ranged from $Y_{NO3} \approx 5.9$ to 6.4 % in reasonable agreement with the literature data (0 – 16 %; Table 1). However, comparing the SOA formation from nighttime chemistry with daytime, the yields from NO$_3$ radical chemistry are higher. The SOA yield this study is very close to those that have been reported by Moldanova and Ljungstrom, 2000 ($Y_{NO3} \approx 0.3 - 6.9$ %) and Nah et al., ($Y_{NO3} \approx 1.7 – 3.6$ %). Although the values agree very well to the majority of the studies, it is still unclear why Fry and co-workers reported no SOA formation from $\alpha$-pinene/NO$_3$ in the presence of seed particles (Fry et al., 2014). Even so, small SOA yields were observed within our present investigation, $\alpha$-pinene/NO$_3$ yielded always SOA. The initial conditions in the study of Fry et al., and this study are very similar, with the exception of the work flow. Fry and co-workers injected the BVOCs into a chamber that was filled with NO$_3$ radicals, whereas for the present study the BVOC was injected at first and afterwards the reaction was initialized. Further studies are needed to reveal the reasons for the discrepancies in the SOA yields from NO$_3$-radical reaction.

Furthermore, comparing the growth curves for OH and NO$_3$ radical reaction with $\alpha$-pinene and limonene, a clear difference in the curve shapes can be seen (Figure 2). The SOA formation from the OH-radical initiated reaction starts later than in the case of NO$_3$ for both systems, $\alpha$-pinene and limonene. Such a long induction period is most likely caused by further reaction of first-generation oxidation products leading to SOA formation as it was demonstrated in previous studies (Ng et al., 2006, Mutzel et al., 2016). As it has been reported by Mutzel et al., 2016, the SOA formation of $\alpha$-pinene /OH and limonene/OH is partly controlled via further reaction of myrtenal and limonaketone/endolim, respectively. The reaction of these first-generation oxidation products is the limiting factor for SOA formation and explains the delay in SOA growth (Mutzel et al., 2016). In contrast, SOA originating from NO$_3$ starts immediately after 30 µg m$^{-3}$ are consumed. Thus, condensable oxidation products are directly formed and partitioned into the particle phase. Potential candidates of those products might be organonitrates.

*Limonene*

Limonene for both oxidation regimes (day and night) yielded the highest SOA yields, compared to α-pinene and

*m*-cresol. The SOA yield from limonene ($Y_{NO3} \approx 16 - 29$ %) is by a factor of 3 - 5 higher than α-pinene and by a factor of 10 higher than *m*-cresol. Those values are close to the lowest values reported for limonene ozonolysis (Northcross and Jang, 2007, Chen and Hopke, 2010, Gong et al., 2018). Furthermore, according to the present data set, limonene with $NO_3$ ($Y_{NO3} \approx 16 - 29$ %) is more efficient in SOA production than the OH radical reaction ($Y_{OH} \approx 10 - 21$ %).

Consequently, nocturnal oxidation of limonene with $NO_3$ yields more SOA than ozonolysis and OH radical reaction. This additional SOA source should be considered in future studies, in particular under less humid conditions. In addition to the strong SOA formation potential, the organic mass production of limonene + $NO_3$ seems to be highly dependent on humidity. This will be discussed separately in the corresponding section below.

*m-Cresol*

In contrast to α-pinene and limonene, *m*-cresol yielded only negligible amounts of SOA with $NO_3$ radicals while the OH radical reaction seems to be more efficient than α-pinene. This observation was unexpected because anthropogenic VOCs are often suggested to form less SOA than biogenic ones. SOA production from anthropogenic VOCs has often been investigated but usually led to inconsistent results and very low yields (e.g. Izumi and Fukuyama 1990, Healy et al., 2009, Emanuelsson et al., 2013). A study by Hildebrandt et al., 2009

raised the question about the low SOA yields and observed much higher yields by using artificial sunlight. The present study also demonstrates higher SOA yields than expected and supports the hypothesis of Hildebrandt and co-workers about a higher importance of SOA production from anthropogenic VOCs.

It should be noted that due to the low SOA yields from $NO_3$ radical reaction, no parameterization of the yield curves can be provided (Figure 1). In general, the SOA yields ($Y_{OH} \approx 2.9 - 9.1$ %) for OH radical reaction with

*m*-cresol are in good agreement to Iinuma and co-workers ($Y_{OH} \approx 4.9$ %), although the photolysis of methylnitrite was used to generate OH radicals. Compared to the study by Nakao et al., 2011 ($Y_{OH} \approx 35 - 49$%) the values are much lower. This is not surprising, as Nakao et al., conducted experiments in the absence of $NO_x$, whereas this study $NO_x$ was always present. The effect of $NO_x$ lowering SOA yields has often been described in the literature (e.g. Presto et al., 2005, Ng et al., 2007, Zhao et al., 2018).

Furthermore, the yield curves clearly indicated a strong effect of relative humidity which can also be seen from the growth curves (Figure 2). This effect will be discussed in the following section.

**3.3 Influence of RH on SOA yield and growth**

Within this study, experiments were conducted at RH <5 %, at 50% and at 75% to investigate the effect of humidity on SOA yield, growth and composition. As discussed in the section above, relative humidity has been

suggested to influence SOA formation and yield for the $NO_3$-radical initiated reaction of VOCs. The observed humidity dependencies could be caused by four main factors: (i) the uptake of the SOA marker compounds or their precursor compounds change as a function of the experimental conditions; (ii) the formation process of SOA marker compounds is directly affected by the experimental conditions; (iii) further reactions take place within the particle phase, and/or iv) the uptake behavior of the first-generation oxidation products might change

with LWC. It remains a challenge to differentiate between all these factors because the observed dependency is most likely the result of a combination of all three factors. A discussion of factor i) to iii) is provided in the respective sections 3.4 Characterization of particle-phase chemical composition. The influence of RH on the

uptake-behavior of first-generation oxidation products cannot be excluded. At it has been demonstrated in previous studies the uptake coefficient of first-generation oxidation products, in particular carbonyl compounds might depend on RH (Healy et al., 2009)

Furthermore, the limited number of studies investigating the effect of RH on the OH radical reaction often contradicts each other (Cocker et al., 2001, Bonn and Moortgat 2002). Only a very limited number of studies is available investigating the influence of RH on SOA formation originating from VOCs+$NO_3$ – the only ones, to the authors' knowledge, are as follows: Spittler et al., 2006, Fry et al., 2009 and Bonn and Moortgat 2002, Boyd et al., 2015. According to Figure 2 a significant effect can be observed for two systems, i.e. limonene/$NO_3$ and $m$-cresol/OH while $\alpha$-pinene/$NO_3$ and $m$-cresol/$NO_3$ were not affected by RH which is in good agreement with the literature studies (by Bonn and Moortgat 2002, Fry et al., 2009, Boyd et al., 2015). Only Spittler and co-workers observed lower SOA yields under humid conditions (RH 20 % vs. 40 %). Notably, in the case of limonene/$NO_3$, the SOA yield varies by a factor of two between 29.1 % (at RH <5 %) and 14.8 % (at RH 75 %). In the literature, higher SOA yields for other cresol isomers have been reported. Due to the different volatilities of the cresol isomers, different SOA formation potentials are expected (Ramasamy et cl., 2019).

In the case of $m$-cresol/OH the SOA yield increases with humidity by a factor of 5. Thus far, no study has investigated the role of RH on the SOA formation from limonene/$NO_3$ and $m$-cresol/OH and the dataset for $\alpha$-pinene /$NO_3$ is small and inconsistent. Since an effect was only observed for limonene/$NO_3$ and $m$-cresol/OH, these systems will be discussed in more detail.

*Limonene + $NO_3$*

The SOA yield was found to decrease with increasing relative humidity from 29 % down to 14.8 %. This pronounced effect could be a result of a direct effect of RH on the partitioning of condensable products. Organonitrates (ON) are well known oxidation products of VOC with $NO_3$ and are often related to SOA formation and growth (e.g. Day et al., 2010, Rollins et al., 2010, Zaveri et al., 2010). ON are reported to be very prone to hydrolysis which might explain the lower SOA yields under humid conditions (e.g. Darer et al., 2011, Hu et al., 2011, Jacobs et al., 2014, Rindelaub et al., 2015). As a contribution of ON is excluded the formation of that hydrophobic compounds that partition into the organic phase needs to be considered as potential explanation. As depicted in Figure 4 limonene yields the highest fraction of water-insoluble OM. Although this process needs to be considered, it is unlikely that this explains the current observation, Figure 2 clearly indicates a decreasing consumption when RH increases as a potential reason for lower SOA yields at higher RH. One might assume that lower consumptions are caused by an enhanced partitioning of $NO_3$ radicals into the particle phase due to an enhanced aerosol liquid water content (ALWC). This seems to be supported by the quantification of particulate inorganic nitrate as this shows higher fraction in SOA under elevated RH (Table 1). Under dry conditions the ratio of produced organic mass:particulate inorganic nitrate is around 9.65 whereas under elevated RH this ratio is decrease to 4.65. Therefore, a stronger contribution of particulate inorganic $NO_3^-$ can be inferred.

A second aspect to be considered is the contribution of first-generation oxidation products. According to theoretical investigations, endolim is the most favored product formed during limonene + $NO_3$ (Jiang et al., 2009). It could be speculated that endolim reacts faster with $NO_3$ than limonene, scavenging $NO_3$ radicals. As no rate constants are available for endolim with $NO_3$, the values from master chemical mechanism (MCM, version 3.3.1, Jenkin et al., 1997, Saunders et al., 2003) were taken for comparison. For the reaction of endolim with

NO$_3$ (LIMAL) a rate constant of 2.6 x 10$^{-13}$ cm$^3$ molecule$^{-1}$ s$^{-1}$ can be found. According to k values taken from MCM and kinetic studies, limonene with NO$_3$ is by two orders of magnitude (k$_{lim+NO3}$: 1.2 6 x 10$^{-11}$ cm$^3$ molecule$^{-1}$ s$^{-1}$) faster compared to endolim. Consequently, a competition between limonene and the respective

first-generation oxidation product can be excluded.

A last sink of NO$_3$ radicals, is the reaction of RO$_2$ radicals with NO$_3$, as it has been investigated by Boyd et al., 2015 for $\alpha$-pinene. Conducting two different sets of reactions with "RO$_2$+NO$_3$ dominant" and "RO$_2$+HO$_2$ dominant" no effect on SOA yield of $\alpha$-pinene was found. Nevertheless, taking into account that $\alpha$-pinene contains only one double-bond, formed RO$_2$ radicals are saturated, whereas limonene as a diene forms RO$_2$

radicals with one remaining double bond, which could be expected to be more reactive than saturated RO$_2$ radicals. Therefore, limonene-originated RO$_2$ radicals are more reactive and might represent an important sink for NO$_3$ which is in competition to limonene + NO$_3$. Thus-far only one study exists investigating this reaction channel. Thus, this competitor for limonene with NO$_3$ seems to be likely and should be systematically investigated in the future.

*m-Cresol+OH*

In contrast to limonene/NO$_3$ the OH radical initiated oxidation of *m*-cresol showed higher SOA yields with increasing RH (Figure 1). Even though the consumption also decreases under humid conditions, the particulate OM is increasing (Figure 2). Analyzing the respective growth curves, a delay in aerosol production can be seen. At RH 0 % and 50 % conditions, aerosol production starts at $\Delta$HC $\approx$ 80 - 90 $\mu$g m$^{-3}$. At more elevated RH the

SOA production starts immediately after initialization of the reaction ($\Delta$HC $\approx$ 5 - 10 $\mu$g m$^{-3}$). According to Ng et al., 2006, such a difference in mass production can be cause by two reasons. The first being a delay in mass transfer from gas into particle phase and/or the second, condensable products are only formed from second-generation oxidation products and thus the formation of those products is the limiting parameter for SOA formation.

According to a study by Coeur-Tourneur et al., 2006, methyl-1,4-benzoquinone (MBQ) is the most dominating oxidation product with up to 12 % molar yield. MBQ was also detected this study by means of the PTR-TOFMS at *m/z* 123. Nevertheless, the increase of the signal in dependency on consumption does not show a significant effect of RH on the formation (Figure 3). Therefore, a strong contribution due to further reactions of MBQ can be excluded. Thus, the delay might be caused by the effect of relative humidity on the partitioning of

condensable products, such as methyl-nitro-catechol. This hypothesis is supported by the comprehensive characterisation of particle phase as discussed in section 3.4.

**3.4 Characterisation of particle-phase chemical composition**

The filters collected after each experiment were analyzed with regards to their content of organic carbon (OC), water-soluble organic material (WSOM), SOA-bound peroxides and SOA marker compounds. The results are

summarized in Figure 4 - 7.

*Organic carbon and water-soluble organic carbon*

Pre-heated quartz fiber filter were analyzed for OM and WSOM content. Please note, WSOM was determined as non-purgeable organic carbon (NPOM). The obtained results were compared to the increase in organic mass ($\Delta$M) obtained from the SMPS (Figure 4). In general, the values agree, meaning the increase in organic mass

corresponds to organic carbon and secondly, the majority of this mass is water-soluble, except in the limonene experiments.

In general, limonene with 22 – 36% of organic mass is the only precursor showing hints for water-insoluble material. From both systems, limonene/$NO_3$ and limonene/OH, the WSOM fraction ranges between 32 – 50 % and 61 – 78 %, respectively. In the case of limonene/$NO_3$ the fraction of water-soluble carbon decreases

dramatically when the relative humidity is reduced. Only one third (32 %) of $\Delta M$ is composed of water-soluble carbon, although the SOA yield was highest under dry conditions. Thus, under reduced RH more water-insoluble compounds partition into the particle phase leading to an enhanced SOA growth. Potential candidates might be higher-molecular weight compounds which seem to be involved in SOA growth for the $NO_3$ and OH system.

To further investigate the fraction of organic material found in the formed SOA, discussions about SOA bound

peroxides and single compounds are provided in the following sections.

*SOA-bound peroxides*

Organic peroxides in SOA were quantified according to a method published by our laboratory (Mutzel et al., 2013), assuming a molar mass of 300 g $mol^{-1}$ (Figure 5), as is recommended by Docherty et al., 2005, presuming that the majority of organic peroxides are higher-molecular weight compounds (e.g. dimers). Notable, the

assumed molar mass has a significant influence of the calculated amount of SOA-bound peroxides. This might cause some uncertainties. The method applies an iodometric detection by UV/Vis spectroscopy. Although, earlier studies demonstrated that $H_2O_2$ injected into aerosol chamber does not cause artefacts, blank experiments were also conducted to exclude them. The blank run was conducted by injecting seed, $H_2O_2$ and NO. After a reaction time of 90 minutes, filter samples were taken and analysed for their SOA-bound peroxide content. Also these

experiments show no peroxide content. Thus for pure inorganic seed particles an effect of $H_2O_2$ originated from the injected $H_2O_2$ can be excluded. Additionally, mixed organic/inorganic seed particles might be prone for partitioning of injected $H_2O_2$. If the organic content would control the $H_2O_2$ uptake, it could be expected that the detected amount of peroxides is the same for particles containing the same amount of OM. In the case of $\alpha$–pinene/OH the content of OM is almost constant (around 5-6 mg $m^{-3}$) whereby the peroxide constant differs

from 80 – 20%. A contribution of mixed organic/inorganic particles an $H_2O_2$ uptake cannot be excluded, but based on the present data set it can be regarded to be very small.

The fraction of SOA-bound peroxides is always expressed as a fraction of organic mass formed during the experiment and was calculated as follows:


$$m_{Perox} = n_{Perox} * 300 \; g \; mol^{-1} \qquad (Eq.\,2)$$

$$M_{Perox} = \frac{m_{perox}}{V_{sampling}} \qquad (Eq.\,3)$$

$$F_{Perox} = \frac{M_{perox}}{M_{org}} \qquad (Eq.\,4)$$

where

$n_{Perox}$ is amount of substance in µmol (calculated from the iodometric peroxide test)

$m_{Perox}$ is mass of organic peroxides in µg

$M_{Perox}$ is mass concentration of organic peroxides in µg $m^{-3}$

$V_{sampling}$ is sampling volume of the filter in $m^3$

$F_{Perox}$ is mass fraction of SOA-bound peroxides in %

$M_{org}$ is amount of organic mass formed during experiment in $\mu g \ m^{-3}$


Organic peroxides were detected from $\alpha$-pinene and limonene, but not from *m*-cresol. The absence of organic peroxides for *m*-cresol might be a result of the aromatic structure. Notably, in the case of $\alpha$-pinene, organic peroxides were only detected from the OH radical reaction, albeit in very high fractions. This observation was unexpected as $NO_x$ was present in the system, usually suppressing ROOH formation. Considering the reaction of

alkylperoxy radicals ($RO_2$) with hydroperoxy radicals ($HO_2$) is the most important source for ROOH, this source decreases with rising $NO_x$-levels due to the competition with $RO_2$+NO (Presto et al., 2005). Other processes than $RO_2$+$HO_2$ should have yielded organic peroxides and thus, other compounds of peroxidic nature are detected by the applied test. As daytime experiments were performed with $H_2O_2$ as OH source, blank filters were carefully checked to exclude the contribution of $H_2O_2$ present in the particle phase due to gas-to-particle partitioning of the

injected oxidant. In general, peroxide fractions of 10 - 80% of the organic mass have been detected from $\alpha$-pinene/OH experiments. The high peroxide fractions of 10 – 80% are in contradiction to the small SOA yields from $\alpha$-pinene/OH ($Y_{OH} \approx 3.5$ %). While the organic peroxide formation from the ozonolysis of $\alpha$-pinene and limonene has been studied in the past (Docherty et al., 2005, Mertes et al., 2012, Epstein et al., 2014, Krapf et al., 2016, Gong et al., 2018), peroxide fractions from the OH-radical induced oxidation are rare (Mertes et al.,

2012). Mertes and co-workers reported peroxide fractions between 5 – 17 % (low $NO_x$ at RH 50 %) and 5.5 - 6.4 % (high $NO_x$ at RH 75 %). Those values are slightly lower than observed in this study with 34 % (medium $NO_x$ at RH 50 %) and 13 % (medium $NO_x$ at RH 75 %). The difference between both studies are most likely caused by usage of other OH radical sources. The observed tendency of lower peroxide fractions under elevated RH is consistent in both studies and might be a result of two facts, i) the uptake of $HO_2$ radicals from the gas phase and

ii) decomposition and/or hydrolysis of hydroperoxides. It has been reported that the gas phase $HO_2$ radical concentration is significantly suppressed by three orders of magnitude when a liquid phase is present (Herrmann et al., 1999). If the $HO_2$ uptake is increased under elevated RH, then $HO_2$, in the gas phase, is only available to a lesser amount to react with $RO_2$ radicals (Herrmann et al., 1999). Furthermore, decomposition and/or hydrolysis occur to a larger extent, lowering the peroxide fraction under high RH (Chen et al., 2011, Wang et al., 2011).

In contrast, limonene yielded SOA-bound peroxides from both oxidation regimes, $NO_3$ and OH. For the OH radical reaction no difference between the respective fractions can be observed, which leads to an average organic peroxide fraction of about 30 % without a dependency on RH. Therefore indicating organic peroxides are i) of a different nature than formed from $\alpha$-pinene, ii) they originate from other reactions and/or iii) the high fraction of water-insoluble material a separate organic phase might be formed protecting peroxide from

hydrolysis. The almost stable content might indicate that those peroxides and their respective formation pathways are not affected by humidity. Thus, it could be speculated that peroxides of higher molecular weight, i.e. dimers with peroxyhemiacetal structure are formed. Due to the lack of data, no comparison to other studies can be provided.

The reaction of limonene with $NO_3$ radicals yielded peroxides as well, in fractions comparable to those measured

for both OH reactions with lower RH. Under higher RH (75 %), the peroxide fraction decreases dramatically to almost 0.5 %. The formation of organic peroxides in $NO_3$-initiated reactions has not been a subject of VOC oxidation studies , with regards to the published information at present. Only a few studies examined the furthur

processing of nitrooxy-, alkyl-, peroxy-radical formed during $NO_3$-radical reaction. For isoprene it has been shown that the reaction of nitrooxy-, alkyl-, peroxy-radical with other $RO_2$ and $HO_2$ is able to form peroxidic
compounds, such as ROOR $C_{10}$-dimers and nitrooxyhydroperoxide (Kwan et al., 2012, Schwantes et al., 2015). As the detected fraction of SOA-bound peroxides decreases with RH, the reaction of $RO_2 + HO_2$ seems to be the major source of these peroxides.

*Biogenic SOA marker compounds*

The quantification of biogenic marker compounds was performed using a BSOA standard containing norpinonic
acid, terpenylic acid, pinonic acid, pinic acid and MBTCA (Figure 6). These compounds were detected from night- and daytime chemistry of α-pinene, while, due to its different structure, limonene does not form these compounds. Therefore, the discussion about the fraction of those BSOA marker compounds refers to those formed within the α-pinene oxidations. Fractions of SOA marker compounds were expressed as mass fraction in formed organic mass (derived from SMPS measurements).

The most important SOA marker compounds formed from the OH-radical reaction of α-pinene are pinonic acid < pinic acid < norpinonic acid and terpenylic acid < MBTCA. A significant dependency on RH was observed in the same manner as with the organic peroxides (Figure 5) for all compounds: Increasing RH leads to decreasing SOA fractional contributions while the SOA yields themselves remain about constant (Figure 1).

As stated above, this observation might be caused by four main factors: (i) the uptake of the SOA marker
compounds or their precursor compounds change as a function of the experimental conditions; (ii) the formation process of SOA marker compounds is directly affected by the experimental conditions; (iii) further reactions take place within the particle phase, iv) the uptake behavior of the first-generation oxidation products might change with LWC. However, it should be noted that earlier studies (Seinfeld et al., 2001, Ma et al., 2007) suggest the aerosol liquid water content (ALWC) does influence the partitioning behaviour of carboxylic acids such as pinic
acid and pinonic acid. As a higher RH corresponds to a higher ALWC, enhanced partitioning is expected. However, this hypothesis is not supported by the experimental results of this study due to the observation of higher fractions of the SOA marker compounds under almost dry conditions, factor i) seems to be unlikely. Factor (ii) may play a partial role in the formation of carboxylic acids. In the literature, the formation of carboxylic acids is often described by OH attacks on a particular precursor compound and subsequent
mechanisms usually involve the reaction of a formed acylperoxy radical with $HO_2$ radicals (Niki et al., 1985, Moortgat et al., 1989, Lightfoot et al., 1992, Larsen et al., 2001). For both reactions, water is most likely to have a direct effect on OH and $HO_2$ radicals, except in opposite directions. Vöhringer-Martinez et al., 2007, suggest that water molecules catalyse the attack of OH radicals (in the gas and particle phases) due to the formation of hydrogen bonds which can lower the reaction barrier. This catalytic effect could lead to higher fractions of
specific markers under elevated RH, which was not observed in the dataset obtained from this study. On the contrary, the reaction of the acylperoxy radical with $HO_2$ might be RH-dependent because, as discussed before, $HO_2$ tends to partition into the particle phase at elevated RH and thus is not sufficiently available in the gas phase (Herrmann et al., 1999). This might explain the low fractions of marker compounds at elevated RH.

The last factor iii) to be considered involves further reactions of the SOA marker compounds to yield high
molecular weight compounds (HMWCs) in the particle phase, as it has been often described in the literature (e.g. Gao et al., 2004, Tolocka et al., 2004, Müller et al., 2008, Yasmeen et al., 2010). It was suggested that compounds such as terpenylic acid and pinic acid can react further in the particle phase to form dimers (Yasmeen

et al., 2010). Thus, under elevated RH, the formation of higher-molecular weight compounds might be enhanced lowering the fraction of individual monomeric compounds. Nevertheless, based on the experimental results of this study and the literature data, the combination of two factors appears to be important for the formation of carboxylic acids. First the suppression of carboxylic acid formation due to enhanced partitioning of $HO_2$ into the particle phase under elevated RH and second, the further reaction of particulate marker compounds yielding HMWCs.

Notably, pinonic acid was detected in comparable amounts from $\alpha$-pinene/OH and $\alpha$-pinene/$NO_3$ with the same RH dependency (Figure 6). The reasons for such a pronounced RH dependency have been discussed intensively above. In addition to the described factors, in the case of $NO_3$-radical reactions, the central role of organonitrates needs to be considered. It can not be ignored that ON act as potential precursors for the detected marker compounds. With elevated RH-conditions and depending on their structure, ON are prone to undergo hydrolysis and the respective products should show higher fractions under elevated RH. This observation seems to be the case for norpinonic acid, leading to the theory these might be the hydrolysis products of respective organonitrates. More data is needed for the hypothesis of norpinonic acid being a hydrolysis product of ON, as it remains speculative at this time.

The remaining marker compounds observed after the $NO_3$ reactions, terpenylic acid, pinic acid and MBTCA, do not show a significant variation with RH and are not affected by water or ALWC. Nevertheless, the fractions of all compounds, except pinonic acid, are significantly lower compared to the OH radical reaction. This is most likely caused by an enhanced formation of ON.

*Anthropogenic SOA marker compounds*

A well-established analytical method was applied to identify and quantify ASOA marker compounds from *m*-cresol oxidation (Hoffmann et al., 2007). Despite the larger number of standards present in the authentic standard mixture, only 3-methyl-6-nitro-catechol (MNC), could be detected and quantified (Figure 7). This compound was only detected from the OH-radical reaction. Quantification of marker compounds in samples after nighttime processing cannot be provided due to the very small SOA yields.

For MNC, an intense RH dependency was found in higher values under humid conditions and lower values under dry conditions (1.5 %). In particular, with elevated RH the fraction of 3-methyl-6-nitro-catechol reached 6 % of overall formed SOA mass, highlighting the importance of this particular oxidation product. Methyl-nitro-catechols (MNC) are of special interest because they are important biomass burning tracer compounds and their ambient concentration can reach up to 29 ng m$^{-3}$ (Iinuma et al., 2010). Additionally, such a high fraction of MNC could also affect the phase state of the particles. It has been shown that MNC particles adsorb water under elevated RH, leading to a change in the phase state of the particles (Slade and Knopf, 2014). This effect is connected to a lower uptake of OH radicals into the particle phase by a factor of 4 when RH increases from 15 to 30 %. The lower fractions under dry conditions might be a result of a stronger OH uptake into the particle phase, leading to a greater extent of heterogenous reactions of MNC in the particle phase.

**3.5 Atmospheric implications and Conclusion**

The examination of the oxidation of atmospherically relevant compounds and the resulting SOA formation is of large importance for a better understanding of atmospheric chemistry, and its response to future climate change. Several studies predict an increase in BVOC emissions as a response to a warmer climate (e.g. Sanderson et al.,

2003, Lathière et al., 2005, Heald et al., 2008). The increase in monoterpenes emission is estimated to be up to 50 % (Lathière et al., 2005). Such a dramatic increase might lead to an enhanced formation of reactive oxygen species (ROS) as determined here as peroxides and SOA. ROS are suggested to cause oxidative stress that influences human morbidity and mortality (Squadrito et al., 2001, Schwartz et al., 2002, Xiao et al., 2003, Ayres et al., 2008). The current knowledge is limited with large uncertainties to predict the global SOA burden (see review by Hallquist et al., 2009, Glasius and Goldstein 2016, Shrivastava et al., 2017). With an expected increase in VOC emissions the knowledge about SOA formation process and their response to changes in the parameters investigated in this study will become more important. The present study provides important data concerning SOA formation potential of OH and $NO_3$ radical oxidation of biogenic and anthropogenic VOCs, the influence of relative humidity on the SOA yield and its resulting chemical composition.

During the night and early morning hours, the RH near the surface is high, $NO_3$ radical chemistry is competitive with that of other oxidants and, accordingly, RH starts to play a crucial role. The investigation of the effect of RH on the SOA formation and chemical composition shed light on various aspects, especially for $NO_3$-initiated SOA formation. $NO_3$ radical reaction can form SOA more efficiently than OH radical reaction in the presence of $NO_x$, at least for $\alpha$-pinene and limonene, highlighting the importance of this atmospherically relevant nighttime sink. Furthermore, pinonic acid was found to contribute significantly, up to 4 %, for $NO_3$- and OH-originated $\alpha$-pinene-SOA, indicating this compound might play a key role for both, day- and nighttime, conditions and not just daytime. Pinonic acid might be formed from the oxidation of pinonaldehyde that itself has been found for $NO_3$-radical initiated reactions of $\alpha$-pinene (Spittler et al., 2006). It should be also noted that huge amounts of organic peroxides were found from $\alpha$-pinene/OH which are an important part of ROS and can be associated with oxidative stress after inhalation of such particles. The peroxide fraction was found to be higher under dry conditions, and, somewhat surprisingly, decreases with RH.

Relative humidity was found to affect SOA growth and composition, in particular the formation of MNC during *m*-cresol oxidation. While daytime chemistry of $\alpha$-pinene and limonene is RH independent ($Y_{OH} \approx 6$ and 20 %), SOA yields from *m*-cresol + OH increased with elevated RH ($Y_{OH} \approx 3 - 9$ %). This observed effect is most likely to be attributed to the huge fraction of MNC with up to 6% under high RH, lowering the uptake of OH radicals and changing the phase state. Additionally, the reaction of limonene + $NO_3$ showed to be very sensitive towards RH, yielding the highest SOA ($Y_{NO3} \approx 29$ %) under dry conditions. This observation is suggested to be caused by a competitive reaction between limonene and formed $RO_2$ radicals, leading to a lower conversion of limonene. Furthermore, *m*-cresol was found to yield only insignificant amounts with $NO_3$, thus producing a highly reactive gas phase, since almost all oxidation products stay in the gas phase. The concentration of reactive species in the gas phase could act as a reservoir for compounds with a much higher SOA formation potential.

*Data availability.* All data presented in this study are available from the authors upon request (herrmann@tropos.de). In addition data from experiment at RH=50% ($\alpha$-pinene/OH, $\alpha$-pinene/NO$_3$, limonene/OH, limonene/NO$_3$, cresol/NO$_3$) are available at the EUROCHAMP webpage (https://data.eurochamp.org/).

*Authors contribution.* AM, OB, HH have planned the experiments. AM, YZ, MR, AK, OB have performed the

chamber experiments. AM, YZ, AK have analysed the data. AM and HH have written the manuscript. YZ, MR, AK, XW, and HH have edited the manuscript.

*Competing interests.* The authors declare that they have no conflict of interest.

*Special Issue statement.*

**5 Acknowledgment**

The present study was supported by the German Research Foundation DFG under grant number HE 3086/25-1. It received further support through funding from the European Union's Horizon 2020 research and innovation programme through the EUROCHAMP-2020 Infrastructure Activity under grant agreement No 730997. Exchange of staff was supported through the EU Marie Skłodowska-Curie Actions, AMIS (295132) and MARSU (690958-MARSU-RISE-2015): All support received is gratefully acknowledged.

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

**Table 1. Experiments conducted for the NO₃- and OH-radical initiated oxidation of α-pinene, limonene and *m*-cresol. All reactions were conducted with 60 ppbv initial hydrocarbon concentration, at T = 293 K and in the presence of (NH₄)₂SO₄/H₂SO₄ (pH = 4 at 50% RH). OH-radical experiments were done in the presence of 10 ppbv NO.**

| Precursor compound | | RH [%] | $\Delta$HC [$\mu$g m$^{-3}$] | $\Delta$M [$\mu$g m$^{-3}$] | SOA yield [%] | NO₃⁻ [$\mu$g m$^{-3}$] | Literature reference[a] |
|---|---|---|---|---|---|---|---|
| α-pinene | NO₃ | <5 | 154 | 10 | 6.7 | 0.30 | 0.2 - 16% (Hallquist et al., 1999); 4 or 16% (Spittler et al., 2006); 1.7 – 3.6% (Nah et al., 2016); 0% (Fry et al., 2014); 9% (Perraud et al., 2010); 0.3 - 6.9 % (Moldanova and Ljungstrom 2000) |
| | NO₃ | 50 | 125 | 7 | 5.9 | | |
| | NO₃ | 75 | 129 | 8 | 6.4 | 0.27 | |
| | OH | <5 | 126 | 5 | 4.1 | | 21.2 % (Ng et al., 2007) |
| | OH | 50 | 115 | 6 | 3.4 | | |
| | OH | 75 | 139 | 6 | 4.3 | | |
| limonene | NO₃ | <5 | 193 | 59 | 29.9 | 3.1 | 14 - 24 % (Moldanova and Ljungstrom 2000); 21 – 40% (Spittler et al., 2006); 25 – 40% Fry et al., 2011); 44 – 57% (Fry et al., 2014 |
| | NO₃ | 50 | 156 | 41 | 26.1 | | |
| | NO₃ | 75 | 107 | 16 | 14.8 | 3.5 | |
| | OH | <5 | 196 | 20 | 10.0 | | 4.3% (Larsen et al., 2001) |
| | OH | 50 | 236 | 50 | 21.0 | | |
| | OH | 75 | 240 | 40 | 19.6 | | |
| *m*-cresol | NO₃ | <5 | 115 | <1 | <1 | < DL | 4.9 % (Iinuma et al., 2010)[b]; |
| | NO₃ | 50 | 102 | <1 | 1.0 | | |
| | NO₃ | 75 | 100 | <1 | 1.7 | < DL | |
| | OH | <5 | 133 | 4 | 2.9 | | 35 – 49% (Nakao et al., 2011) |
| | OH | 50 | 114 | 8 | 6.2 | | |
| | OH | 75 | 84 | 10 | 9.1 | | |
| Blanks | NO₃ | 50 | - | <1 | <1 | | |
| | OH | 50 | - | <1 | <1 | | |

[a]only those studies are reported for OH radical reaction of limonene and α-pinene that apply also H₂O₂/NO as OH source; [b] due to the lack of data all available literature is shown

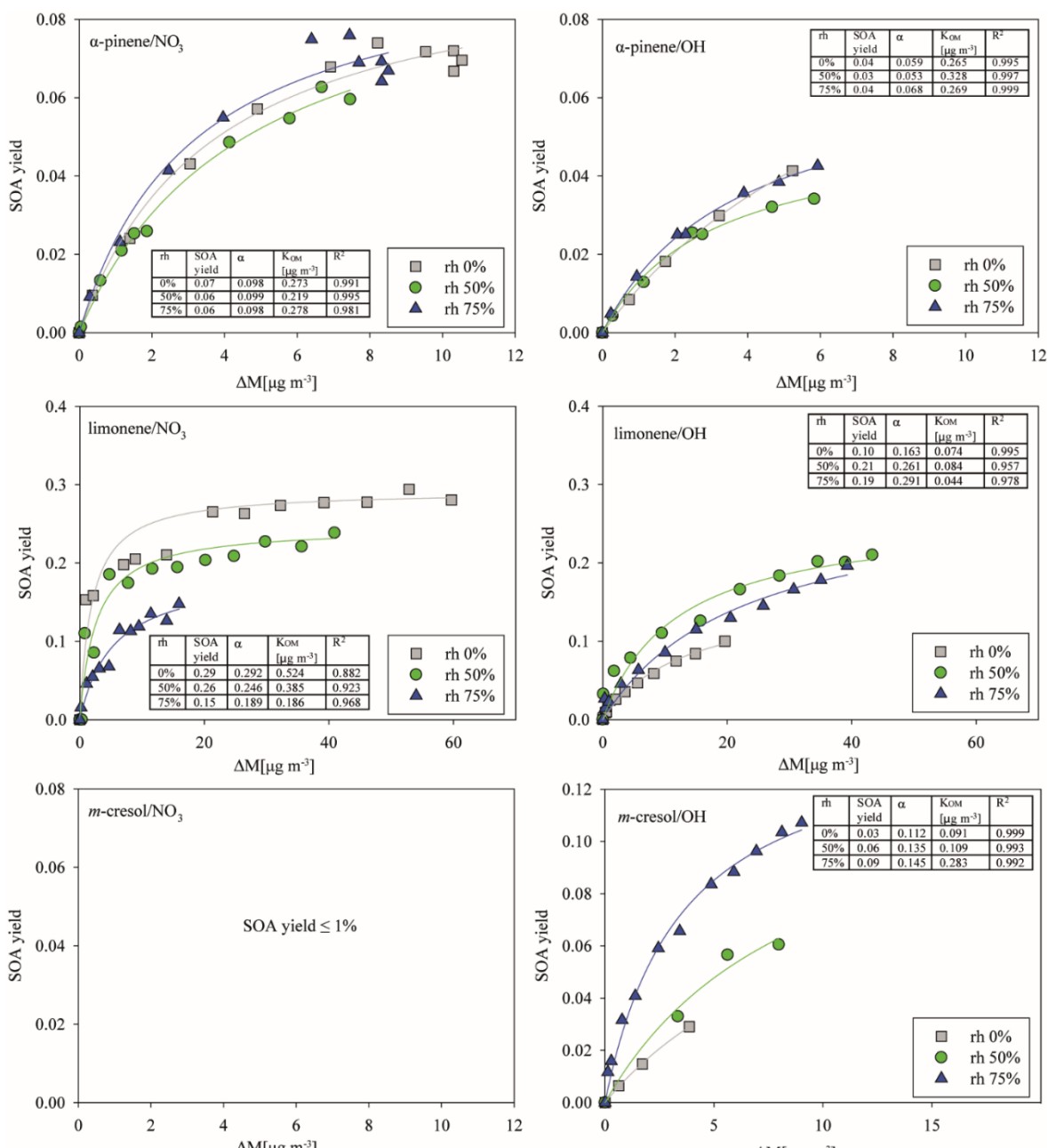

Figure 1. Yield curves for α-pinene, limonene and *m*-cresol with NO₃ and OH radicals for RH 0%, 50% and 75%. Yield curves were parametrised with the one-product approach (Odum et al., 1996). The obtained values for α (mass yield) and K$_{OM}$ (partitioning coefficient) are included as well. Please note, the SOA yield of m-cresol/NO₃ was below 0.01%. Therefore, no parameterisation can be provided. Each fit present a single chamber experiment. SOA yield was calculated by deviating the produced organic mass by the consumed amount of hydrocarbon.

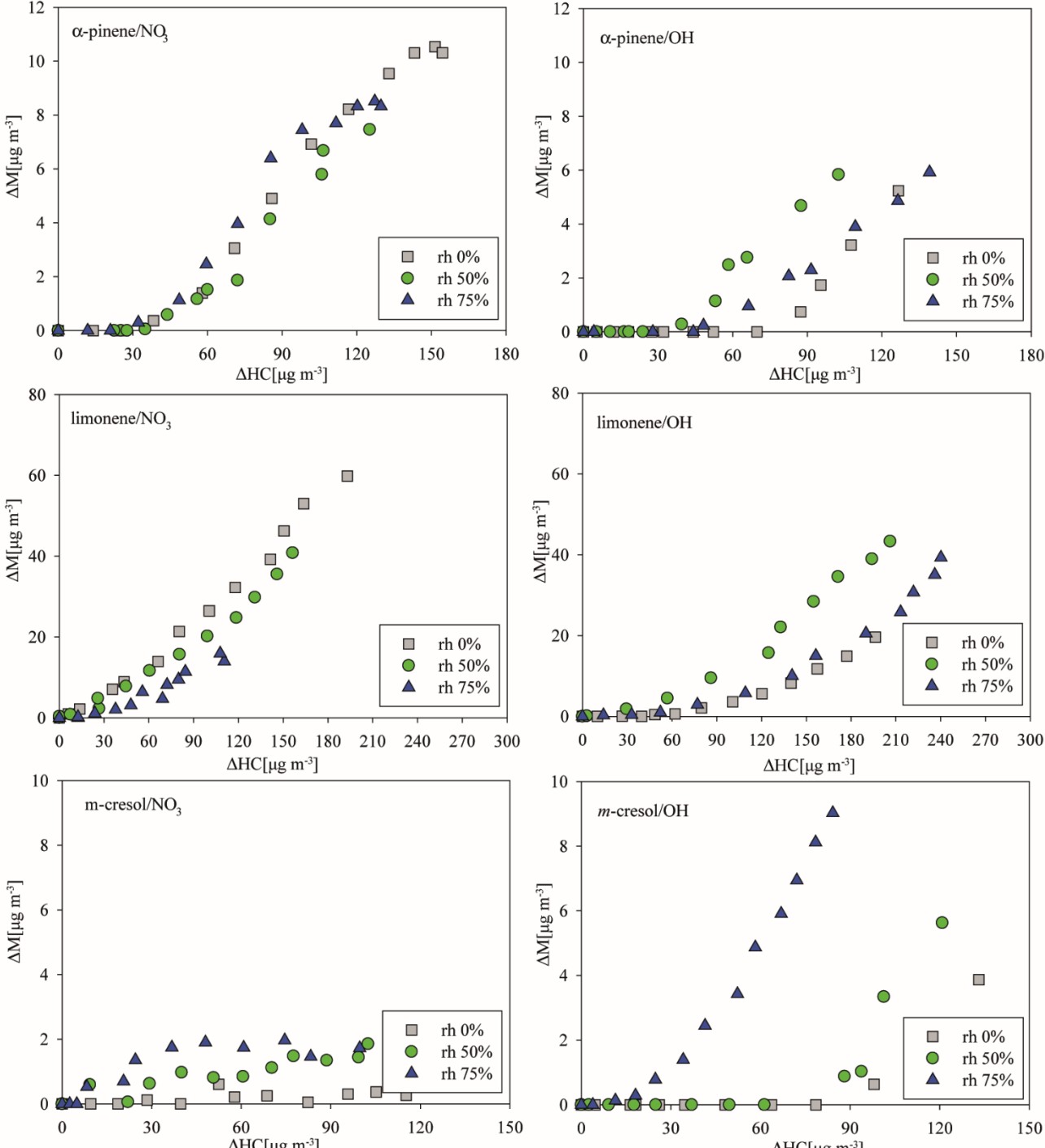


**Figure 2.** Growth curves for α-pinene, limonene and *m*-cresol with NO₃ and OH radicals for RH 0%, 50% and 75%.

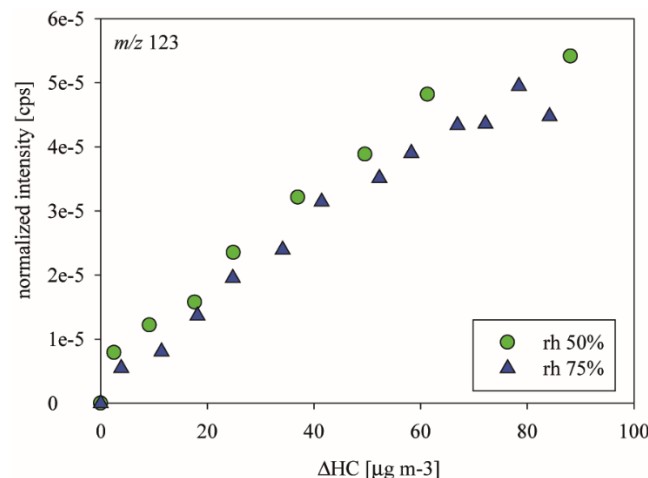

**Figure 3.** Evolution of *m/z* 123 as a function of consumption of *m*-cresol (ΔHC). The signal at m/z 123 can be attributed to methyl-960  1,4-benzoquinone.

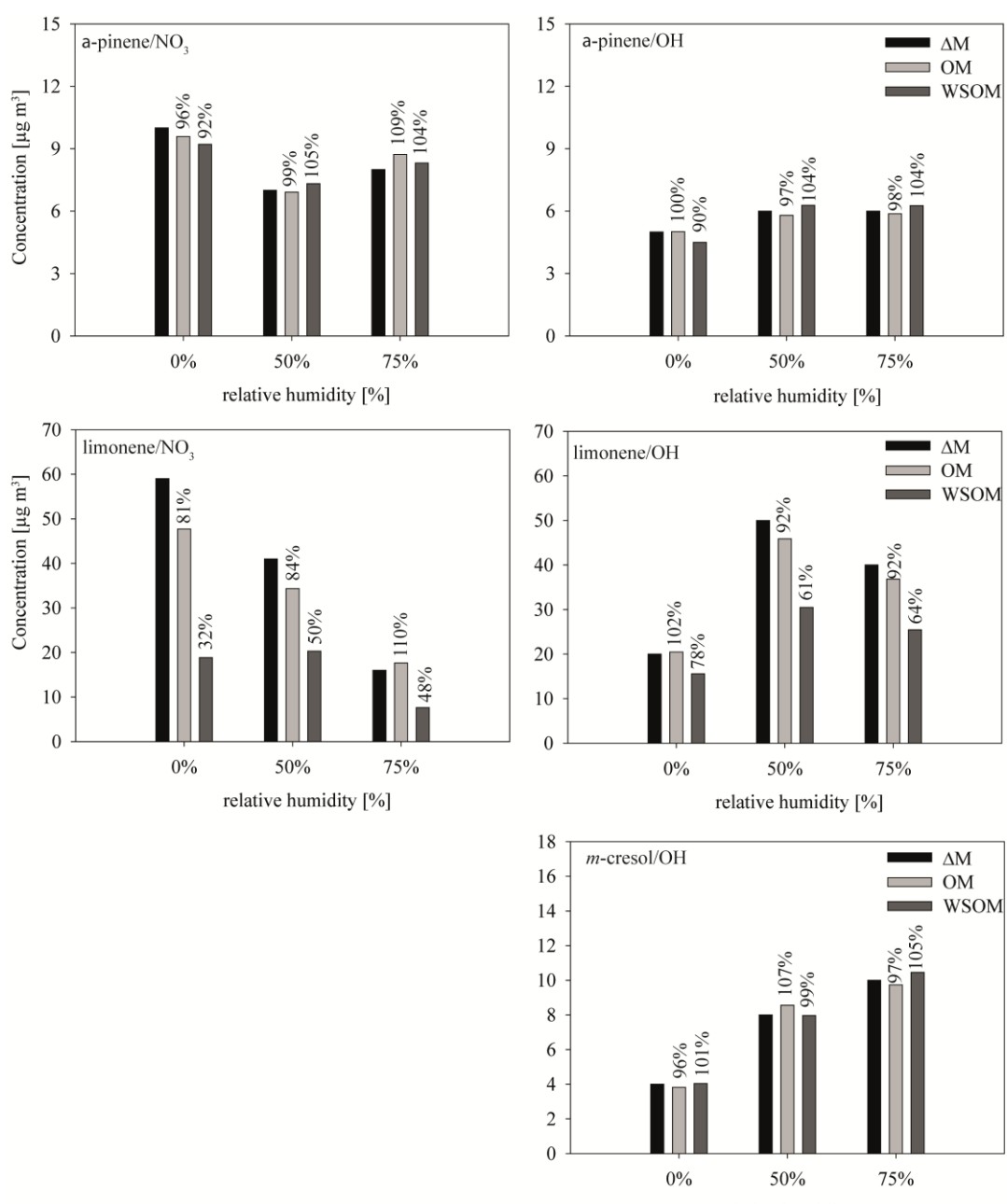

**Figure 4. Comparison of organic mass calculated from SMPS with an offline determined concentration of organic material (OM) and non-purgeable organic material (OM). Measurement uncertainties are given as follows 10% for SMPS measurements (Wiedensohler et al., 2012), 5% for OC/EC measurements (Spindler et al., 2004) and 10% for WSOM measurements (Timonen et al., 2010). Please note, no values can be given for m-cresol/NO₃ as the produced organic mass was too small to be within the detection limits of the different techniques. The values of OM and WSOM were also illustrated as fraction of the produced organic mass ΔM (expressed as % above to the corresponding bar).**


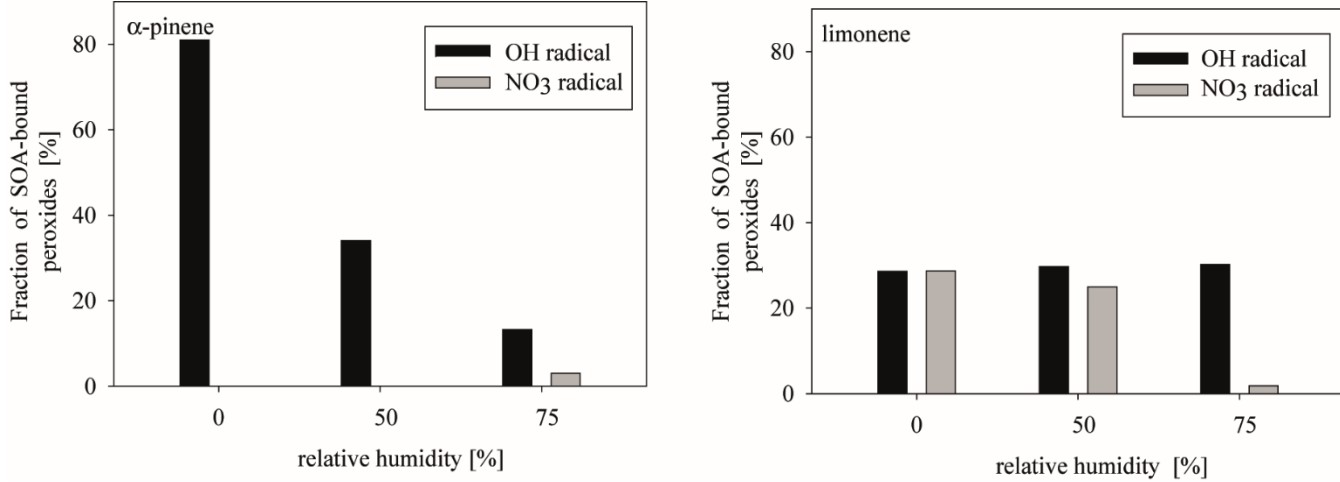

**Figure 5. Fraction of SOA-bound peroxides from α-pinene (left) and limonene oxidation with OH radicals (black) and NO₃ radicals (grey). Quantification was done following the method described by Mutzel et al., 2013 assuming a molar mass of 300 g mol⁻¹ (Docherty et al., 2005).**

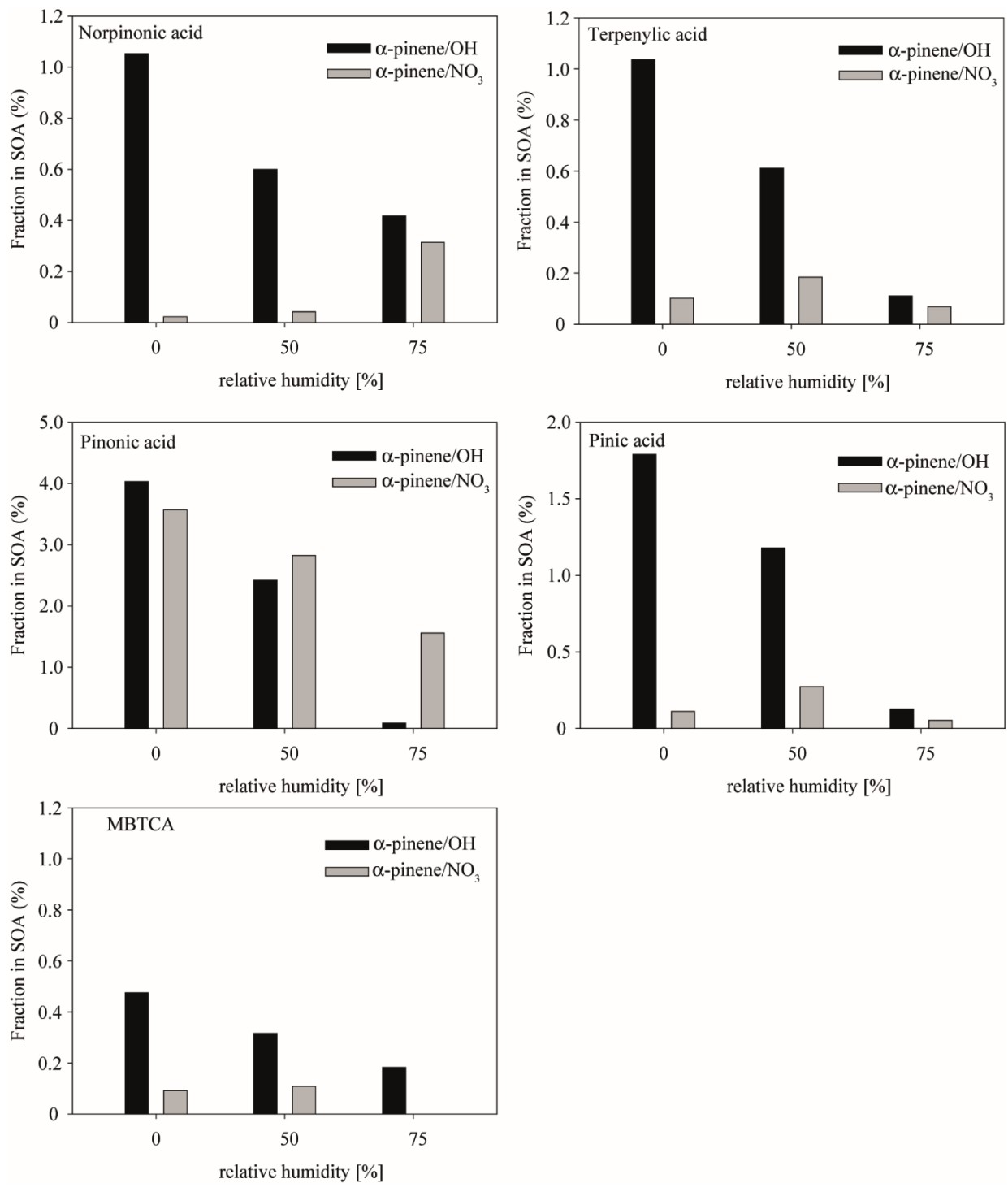

**Figure 6. Fraction of α-pinene marker compounds norpinonic acid, terpenylic acid, pinonic acid, pinic acid and MBTCA for OH-radical reaction (black) and NO₃-radical reaction (grey).**

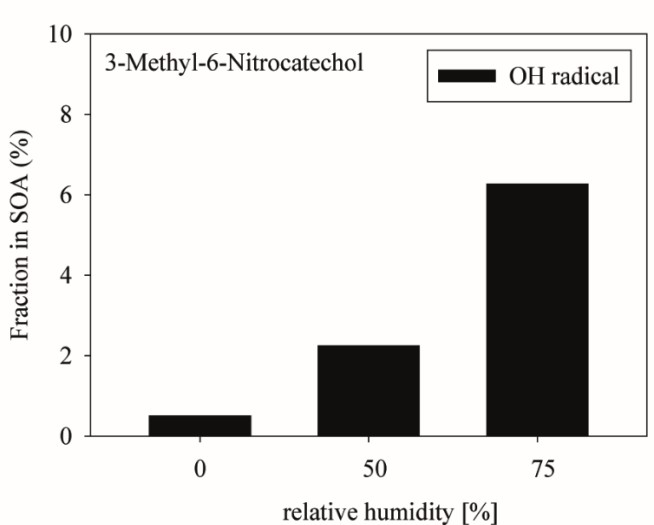

**Figure 7. Fraction of 3-methyl-6-nitrocatechol from the oxidation of *m*-cresol with OH. Please note, other compounds from the ASOA standard (Hoffmann et al., 2007) were not identified and due to the low SOA yield from NO$_3$-radical reaction, the concentration might be below the detection limit.**