# Peer review of "Importance of SOA formation of $\alpha$ -pinene, limonene and *m*-cresol comparing day- and nighttime radical chemistry"

_Atmospheric Chemistry and Physics, 2019_

## Referee Comment (RC1) · Anonymous Referee #1 · 2 Mar 2020

**Review: Importance of SOA formation of a-pinene, limonene and m-cresol comparing day- and night-time radical chemistry. Mutzel et al., acp-2019-1066.**

This paper describes an experimental study looking at the SOA formation potential of these three important species with OH radicals in the presence of NO and from nitrate radical chemistry. It also investigates the effect of changing the RH in the chamber on the SOA yield and on the yield of individual SOA tracer species. The experiments and the premise of this paper are very interesting and could provide valuable information for the community. However, I felt that some of the analysis was rather limited and hypothesis were presented without evidence, even though the types of analysis used, could have provided this. In addition, some of the language was hard to follow and could benefit from being tidied up. I think if the major points below are addressed the paper would be suitable for ACP.

Major comments

Page 8, line 260-268. I do not agree with the reasoning that the inorganic nitrate is not higher under humid conditions. At <5 % RH, the ratio of DM:NO3- is 29.9:3.1 = 9.65. In contrast, at 75 % RH, the ratio is 16:3.5 = 4.65. A much larger fraction of the aerosol produced is nitrate at higher RH. Also, if any organic nitrates that are present in the SOA do hydrolyse, then they could appear as $NO_3^-$ in the IC analysis, since this uses a water extraction step.

Page 10, line 340-345: What were the results of the blank tests? The a-pinene results are indeed very unexpected. Can you estimate the proportion of $RO_2$ reacting with $HO_2$ versus NO in these experiments? What is the temporal evolution of NO in the chamber, how quickly is it converted to $NO_2$? Also, how much ozone is formed? Could you be seeing ozonolysis leading to SOA peroxides? Your values seem so much higher than previous studies. Could other species be giving a response? Needs more information in the experimental about the analysis method, not just giving a previous paper. For limonene, surely you could use the MS data to determine if high molecular weight peroxide dimers were present?

Page 11, section on Biogenic SOA markers: There is no comparison of the a-pinene SOA tracer data to previous studies. Are the levels observed in alignment with previous studies (especially when considering such high peroxide content)? There needs to be a better discussion here about the amount of liquid water content present at the different RH levels. Is there really less pinonic acid formed at 75 % RH or is there just more water in the SOA, leading to a more dilute aerosol? Line 406: Some of the dimers can be observed in negative ionisation ESI. Could you qualitatively determine whether this pathway increases at higher RH using the LC-MS data?

Page 12, section on Anthropogenic SOA markers: I don't really understand why you are comparing the composition to an "ASOA marker" mixture, rather than using existing knowledge about cresol chemistry to use appropriate standards of oxidation products. Also, there is discussion here about biomass burning but you have called this an anthropogenic tracer. The Hoffmann paper (2007) calls this a method for biomass burning. How many of these compounds would you expect to form from m-cresol oxidation? Are there species you would have expected to see but don't? While the 3MNC shows a positive trend, I don't really feel that this data adds very much to our understanding of cresol SOA.

Page 13, line 465: Have there been any studies of the OH yield from NO3+a-pinene? The formation route for pinonic acid is not obvious. Do you have any idea where it is coming

from in the NO3 reaction?  Is it similar to the norpinonic and a possible hydrolysis product? Or could it be from residual ozone or OH?

Page 13, line 478: I felt the last few lines made a very important comment that was not really discussed in the main paper.  To me it seems very odd that so little SOA is formed from this route as such a large amount of m-cresol is reacted. In your table 1, there is a literature reference suggesting a 35-49 % yield, but I think perhaps this should be for OH chemistry? A recent paper on o-cresol +NO$_3$ indicated a SOA yield of 7-17%. There needs to be much more discussion about the reasons for this in the main paper on in the section on page 7.

Sathiyamurthi Ramasamy, Tomoki Nakayama, Takashi Imamura, Yu Morino, Yoshizumi Kajii, Kei Sato, Investigation of dark condition nitrate radical- and ozone-initiated aging of toluene secondary organic aerosol: Importance of nitrate radical reactions with phenolic products, Atmospheric Environment, 2019, https://doi.org/10.1016/j.atmosenv.2019.117049

Minor comments
Abstract:  why only give a-pinene peroxide levels here? Also, "pinonic acid as the most important particle phase constituent" What do you mean by "important"? Do you mean highest proportion based on the small subset of products measured?

Page 2, line 57 – is the uncertainty on the limonene + NO$_3$ rate coefficient really a factor or almost 100?  Seems very high.

Page 2 -the introduction is very focussed on NO$_3$ chemistry and very little on OH. There is clearly much more literature information available for OH and I would like to see some of this incorporated here.

Page 2, line 69:  Why is cresol used as the anthropogenic tracer?  No explanation given for this choice, since it can also be released from biomass burning.

Page 3, line 96:  "equals this a dilution" – consider rewording this sentence.

Page 3, line 104:  Can you explain how the pre-reactor was set up to avoid ozone entering the chamber.  How much ozone did you detect in the chamber during NO$_3$ experiments?

Page 4, line 114:  1 g cm$^{-3}$ seems a rather low value to use for density of SOA.

Page 4, line 134:  should this say micro L not milli L?

Page 5, section 2.3.4:  The contents of the anthropogenic SOA mix need to be included here. The reader does not want to have to go and look this up.  It is important to know whether you would expect to see any other tracers in the mix in the cresol experiment or not. If not, why not just use a 3MNC standard?

Page 5, line 164:  Reconsider this sentence.  Im not sure how it can be "significantly or at least slightly higher"?

Page 6, line 190:  This sentence doesn't make sense.  I think you are trying to say the yield from NO3 is higher than from OH.  But this sentence seems to say the opposite. Only one a-pinene + OH reference is given in Table 1. I am sure there are many more than that. Page 6,

Page 6, line 201:  I don't follow this sentence.  Do you mean SOA is being formed from the oxidation of first generation products rather than the products themselves?

Page 6, line 204:  Please give structures for the non-standard named compounds like Myrtenal.

Page 6, line 216 – please give typical limonene ozonolysis SOA yields.

Page 7, line 222 – is there no literature data for SOA from cresol for comparison? The start of this section is rather imprecise. Using cresol as a proxy for "anthropogenic" VOCs and comparing whether they form more SOA than biogenics is not accurate. Hildebrandt uses the parent hydrocarbons and so cannot simply be applied to the cresol.

Page 7, line 240 – there is no mention here about the impact of RH on aerosol phase and hence the uptake of SVOC.

Page 7, line 250: You have not mentioned limonene + OH, which seems to have a different pattern to the others, with the highest yield at 50 % RH?

Page 8, line 277: can you really say this based on a SAR derived rate constant from the MCM?

Page 8, line 280: Were these two sets of reactions done in this study? What is the "one study that investigated this reaction channel"?

Page 8, line 297 – I don't understand what "the evolution in dependency on consumption" means.

Page 9, line 305: This whole section is really confusing, with long sentences with too many parts. There needs to be some more discussion about the data in the figure. Why are some values over 100 %? The proportion of OM also is much lower in limonene + $NO_3$, why is this? I felt there was a lot of information in the figure, but very little discussion of what it means and the levels of uncertainty.

Page 9, line 321: What is the sensitivity of this analysis as a result of changing the molar mass? There is increasing evidence of auto-oxidation in some systems, which can occur in the presence of NO, which could shift the peroxide mass distribution.

Page 10, line 355 to 360: can you make it clear that you are talking about HO2 concentrations in the GAS phase.

Page 11, line 415. I don't follow this sentence. To me the RH dependence looks very different – nowhere near as dramatic at 75 % RH.

---

## Referee Comment (RC2) · Anonymous Referee #2 · 4 May 2020

General comments:

This is a potentially interesting study describing the influence of RH and oxidants on the SOA formation potential of some of most important VOCs. My major concern with this manuscript is that the discussion is unclear and does not provide enough analysis or evidence. Unfortunately, some parts of the manuscript are hard to follow, and this does not help the clarity of the discussion. Nevertheless, the chamber experiments and sample analysis are done well, and the data can be potentially useful to other SOA experimentalists and modelers. The manuscript may be suitable for publication after improving its clarity, and addressing specific comments below.

[Figure]

Interactive
comment

Specific comments:

Please use abbreviation consistently. It improves the readability of the manuscript greatly. E.g. RH and relative humidity.

L23 In average -> On average

L127: Please give the name of an IC-CD model and manufacturer.

L150: Please provide a reason for changing the chromatographic condition.

L186: Do the authors mean "The values obtained in this study did not agree with previously reported values because OH sources are different"?

L239: This should be 3.2. Please change all the section numbers accordingly.

Line 262: I am not too sure what the authors mean here by "a decreasing consumption when RH increases". Do the authors mean "a decreasing consumption of VOC when RH increases"? If so, this seems to contradict what Figures 1 and 2 (limonene/NO3) shows. Please clarify this sentence.

L307: NPOM should be described in line 73.

L315: It may well be that higher molecular weight compounds do not ionize well in negative ESI, and they aren't present in the LC/MS data. Just in case, do the authors have signals for them in LC/MS data?

L346: a-pinene/OH -> $\alpha$-pinene/OH

L346-347: Why do high peroxide fractions contradict the small SOA yields? Please provide the reason for this explanation.

L376 Biogenic SOA marker compounds and Figure 6: Does the discussion take the water content of SOA? If not, it makes more sense to discuss data in terms of carbon mass fraction of marker compounds in OC to eliminate the effect of water content.

L865 Figure 4 Caption: non-purgeable organic material (OM) -> non-purgeable organic

material (NPOM).

L415: Do the authors mean pinonic acid was detected in comparable fractions in both NO3 and OH oxidation of $\alpha$-pinene? If so where do these 20-25% come from? For the 75%RH experiments, $\alpha$-pinene/OH experiment shows much lower pinonic acid fraction. Can the authors clarify this?

————————————————

---

## Author Comment (AC1) · 12 Oct 2020

We gratefully thank the reviewer for the comments. Please find the corresponding answers below. Changes in the manuscript are highlighted in blue.

**Reviewer 1**

*Reviewer comment: Page 8, line 260-268. I do not agree with the reasoning that the inorganic nitrate is not higher under humid conditions. At <5 % RH, the ratio of DM:NO3-is 29.9:3.1 = 9.65. In contrast, at 75 % RH, the ratio is 16:3.5 = 4.65. A much larger fraction of the aerosol produced is nitrate at higher RH. Also, if any organic nitrates that are present in the SOA do hydrolyse, then they could appear as NO3-in the IC analysis, since this uses a water extractionstep.*
Authors response:
We do agree to this comment. The absolute value of nitrate stayed constant under dry and humid conditions. While calculating the ratio of organics to nitrate under both conditions another conclusion can be found. The ratio is much lower under humid conditions indicating a high contribution of nitrate. The manuscript was changed accordingly and the new text now reads:

**Page 8, line 262:** *One might assume that lower consumptions are caused by an enhanced partitioning of $NO_3$ radicals into the particle phase due to an enhanced aerosol liquid water content (ALWL). This seems to be supported by the quantification of particulate inorganic nitrate as this shows a higher fraction in SOA under elevated RH (**Fehler! Verweisquelle konnte nicht gefunden werden.**). Under dry conditions the ratio of produced organic mass:particulate inorganic nitrate is around 9.65 whereas under elevated RH this ratio decreases to 4.65. Therefore, a stronger contribution of particulate inorganic $NO_3^-$ can be suggested.*

*Reviewer comment:Page 10, line 340-345: What were the results of the blank tests? The a-pinene results are indeed very unexpected. Can you estimate the proportion of RO2reacting with HO2versus NO in these experiments? What is the temporal evolution of NO in the chamber, how quickly is it converted to NO2? Also, how much ozone is formed? Could you be seeing ozonolysis leading to SOA peroxides? Your values seem so much higher than previous studies. Could other species be giving a response? Needs more information in the experimental about the analysis method, not just giving a previous paper. For limonene, surely you could use the MS data to determine if high molecular weight peroxide dimers were present?*

Authors response:
The reviewer here addresses important issues. The blank test was conducted to evaluate the formation of SOA by side reactions, in particular inorganic nitrate and to exclude a contribution of H2O2 used for OH-radical generation during the determination of SOA-bound peroxides. As it can be seen no SOA, nitrate could or SOA-bound peroxides could be detected from the blank runs.
An estimation of the formed amount of $RO_2$ and $HO_2$ would be very interesting. As the present study is solely focused on experimental measurements, modeling of the experimentally obtained data might well be a topic of a follow-up manuscript. The discussion on the branching ratio of $RO_2+HO_2$ and $RO_2+NO$ is given at page 10, line 345-352.
Within a follow-up study an AMS could be connected to the chamber to monitor online the particle-phase composition. Unfortunately, AMS was not present during the measurements the results of which are presented here.
We certainly agree on reviewers comment that the fraction of peroxides is high for a-pinene/OH. Nevertheless, they are lower than values reported by Docherty et al., 2005. As the reviewer stated in a later comment, the peroxide level is very sensitive towards molar mass. Using a lower molar mass would

lower the peroxide fraction. Compounds that might interfere with the peroxide test are peroxy acyl nitrates (PAN). A respective paragraph is added to the manuscript, i.e.:

**Page 10, line 352:** *Peroxy acyl nitrates (PAN-type compounds) would be potential candidates or the contribution of $H_2O_2$ partitioned from gas to the particle phase.*
**Page 27, line 887:** *Please note, according to Eq. 1-3 the peroxide fraction is highly sensitive towards the molar mass used for calculation.*

*Reviewer comment: Page 11, section on Biogenic SOA markers: There is no comparison of the a-pinene SOA tracer data to previous studies. Are the levels observed in alignment with previous studies (especially when considering such high peroxide content)? There needs to be a better discussion here about the amount of liquid water content present at the different RH levels. Is there really less pinonic acid formed at 75 % RH or is there just more water in the SOA, leading to a more dilute aerosol?*

Authors response:
To the authors' opinion the discussion on the LWC at page 11 is sufficient. Different routes how water can affect the partitioning behavior and also the formation of carboxylic acids are given. A dilution of the target compounds might be relevant while interpreting absolute concentrations. To avoid those effects, the fraction of marker compounds within the formed SOA is illustrated (Figure 6). While analyzing the mass fraction, effects such as dilution shouldn´t disturb. To highlight this fact, a sentence is added to the manuscript. Furthermore, as only mass fractions are illustrated comparisons to other available literature is not needed. Additionally, the yield and mass fraction highly depends on LWC, phase state, composition of the organic phase etc.

**Page 28, line 892:** *Please note that the fractions of marker compounds in the SOA formed SOA are illustrated.*

*Reviewer comment: Line 406: Some of the dimers can be observed in negative ionisation ESI. Could you qualitatively determine whether this pathway increases at higher RH using the LC-MS data?*
Authors response:
We agree on reviewer comments that a quantification of dimers is missing. Unfortunately, no standards are available and thus a quantification cannot be provided. The synthesis of those standards for quantification would be very beneficiary for the whole aerosol community but is out of scope for the present study. As it is stated in the manuscript, this is a suggestion based on literature findings.

*Page 12, section on Anthropogenic SOA markers: I don't really understand why you are comparing the composition to an "ASOA marker" mixture, rather than using existing knowledge about cresol chemistry to use appropriate standards of oxidation products. Also, there is discussion here about biomass burning but you have called this an anthropogenic tracer. The Hoffmann paper (2007) calls this a method for biomass burning. How many of these compounds would you expect to form from m-cresol oxidation? Are there species you would have expected to see but don't? While the 3MNC shows a positive trend, I don't really feel that this data adds very much to our understanding of cresol SOA.*
Authors response:
We do not agree on reviewer comment. As it was pointed out, cresols are important biomass burning marker compounds are often related to the described ASOA marker mixture. Many of these compounds have been described also in literature to be formed during cresol oxidation. Thus, the used mixture seemed to be representative. Furthermore, it should be clear that the present study was focused solely on SOA formation and thus particulate oxidation products were in the focus of the work. Even though few studies exist on pure cresol-oxidation, the minority of them provides products identification in the

particle phase using authentic standard compounds. 3MNC was the only compounds we could observe with applied technique. In the future other techniques than the one applied in the present study could provide better insight into the particle-phase composition (e.g. IC/MS for small functionalized organic acids)

*Reviewer comment: Page 13, line 465: Have there been any studies of the OH yield from NO3+a-pinene? The formation route for pinonic acidis not obvious. Do you have any idea where it is coming from in the NO3 reaction? Is it similar to the norpinonic and a possible hydrolysis product? Or could it be from residual ozone or OH?*

The formation of pinonaldehyde is described in Spittler et al., 2006 via formation of the nitrate-peroxy-radical. As pinonaldehyde is a potential pinonic acid precursor this pathway could be considered. A corresponding paragraph is added to the manuscript which reads:

**Page 13, line 462:** *The formation of pinonic acid might proceed via further reaction of pinonaldehyde that this described to be formed during NO₃-radical initiated reaction of a-pinene (Spittler et al., 2006).*

*Reviewer comment: Page 13, line 478: I felt the last few lines made a very important comment that was not really discussed in the main paper. To me it seems very odd that so little SOA is formed from this route as such a large amount of m-cresolis reacted. In your table 1, there is a literature reference suggesting a 35-49 % yield, but I think perhaps this should be for OH chemistry? A recent paper on o-cresol +NO3indicated a SOA yield of 7-17%. There needs to be much more discussion about the reasons for this in the main paper on in the section on page 7. Sathiyamurthi Ramasamy, Tomoki Nakayama, Takashi Imamura, Yu Morino, Yoshizumi Kajii, Kei Sato, Investigation of dark condition nitrate radical- and ozone-initiated aging of toluene secondary organic aerosol: Importance of nitrate radical reactions with phenolic products, Atmospheric Environment, 2019, https://doi.org/10.1016/j.atmosenv.2019.11704*

We agree that the reference by Nakao et al. belongs to OH-radical chemistry. Accordingly, we shifted it. We apologize for this mistake.
In fact, the low SOA yield of m-cresol+NO₃ was very unexpected. But as all three cresol isomers have very different physical properties in terms of volatility, it seemed to be reasonable. o-cresol is less volatile than m-cresol, and, accordingly, the corresponding products might also be of lower volatility as well leading much faster to SOA formation. Therefore, both studies are not contradicting each other.
The reference Ramasamy et al. has now been included into the manuscript. Please note that this publication was not included into the orginial manuscript as it was published only after the present work was submitted.

**Page 7, line 251:** *In the literature, higher SOA yields for other cresol isomers have been reported. Due to the different volatilities of the cresol isomers, different SOA formation potentials are expected (Ramasamy et cl., 2019).*

Minor comments:
Abstract: why only give a-pinene peroxide levels here? Also, "pinonic acid as the most important particle phase constituent" What do you mean by "important"? Do you mean highest proportion based on the small subset of products measured?
The fraction of SOA-bound peroxides was added to the abstract. Pinonic acid was highlighted as most important product as it was found with 1-4 % which is a significant fraction for a single compound

The abstract was changed accordingly and this text now reads:

**Page 1, line 28:** *The fraction of SOA-bound peroxides which originated from □-pinene varied between 2 – 80% as a function of RH and was found to be around 30% for limonene/Oh independent of the RH.*
*Furthermore, SOA from α-pinene revealed pinonic acid as the most significant single particle-phase constituent under day- and night-time conditions with a SOA fraction of 1 – 4%.*

Page 2, line 57 –is the uncertainty on the limonene + NO3 rate coefficient really a factor or almost 100? Seems very high.
Yes, the rate constants reported in the literature vary dramatically.

Page 2 -the introduction is very focussed on NO3chemistry and very little on OH. There is clearly much more literature information available for OH and I would like to see some of this incorporated here.
Authors response:
Certainly, more literature is available on OH radical chemistry. But the focus of the present manuscript is $NO_3$-radical chemistry and, accordingly, we highlighted the importance of $NO_3$ radical chemistry.

Page 2, line 69: Why is cresol used as the anthropogenic tracer? No explanation given for this choice, since itcan also be released from biomass burning.
Authors response:
Cresol is an important biomass burning tracer compound and only few studies are available that investigate cresol oxidation. Thus, it was included into the present study. The importance of cresol and the respective night-time chemistry is clearly stated in the atmospheric implication and conclusion section.

Page 3, line 96: "equals this a dilution" –consider rewording this sentence.
The sentence reads now:
Based on a reaction time of 90 minutes a dilution of 4.7 % ($NO_3$) and 2.4 % (OH) can be estimated.

Page 3, line 104: Can you explain how the pre-reactor was set up to avoid ozone entering the chamber. How much ozone did you detect in the chamber during NO3experiments?
The pre-reactor was described and characterised within a previous study (Iinuma et al., 2010.) from our laboratory. A detailed description can be found therein. The paper is now cited in the experimental description. As the pre-reactor was not newly developed for this study, a detailed characterisation was not included.

**Page 3, line 86:** *A fraction of the air flow (10 L min$^{-1}$) out of the total air flow in the flow tube (30 L min$^{-1}$) was directed to the chamber (Iinuma et al., 2010).*

Page 4, line 114: 1 g cm-3 seems a rather low value to use for density of SOA.
As the real density is unknown a standardized density of 1 g cm-3 is used. As we state clearly which density is used, one can use it and convert by factors to be able to compare to studies using another density.

Page 4, line 134: should this say micro L not milli L?
The manuscript was changed accordingly. *Page 4, line 134: 500 uL*

Page 5, section 2.3.4: The contents of the anthropogenic SOA mix need to be included here. The reader does not want to have to go and look this up. It is important to know whether you would expect to see any other tracers in the mix in the cresol experiment or not. If not, why not just use a 3MNC standard?

We agree and a list of compounds being present in the mixture is added now to the manuscript. The text now reads:

**Page 5, line 157:** *The composition of the anthropogenic SOA mix for biomass burning is described in detail in Hoffmann et al., 2007 containing 3,5-dimethoxy-4-hydroxy-acetophenone, 4-hydroxycinnamic acid; Sinapic acid; Ferulic acid; Vanillic acid; Vanillin; Homovanilic acid; Syringic acid; Syringaldehyde; Coniferyl aldehyde; Nitrocatechol; 4-nitrophenol; 4-nitroguaiacol; 2,4-dinitrophenol; 2-nitrophenol; 3-methyl-4-nitrophenol; 2,6-Dinitro-4-methylphenol; 2-methyl-4-nitrophenol; 2-methyl-4,6-dinitrophenol; 4-methyl-2-nitrophenol and 2,6-Dimethyl-4-nitrophenol.*

The used standard is a well-established mixture of atmospheric relevant compounds that are often observed from biomass burning events. Thus, it was worth evaluating if other compounds were formed as well. As this was not the case, no further discussion was provided.

Page 5, line 164: Reconsider this sentence. Im not sure how it can be "significantly or at least slightly higher"?

The manuscript was changed accordingly.

**Page 5, line 168:** *In general, $\alpha$-pinene yielded higher SOA with NO$_3$ radicals ($Y_{NO3} \approx 6\%$) than with OH ($Y_{OH} \approx 3.5$ %).*

Page 6, line 190: This sentence doesn't make sense. I think you are trying to say the yield from NO3 is higher than from OH. But this sentence seems to say the opposite. Only one a-pinene + OH reference is given in Table 1. I am sure there are many more than that.

The manuscript was changed accordingly to:

**Page 6, line 190:** *However, comparing the SOA formation from night-time chemistry with day-time, the yields from NO3 radical chemistry are higher.*

Yes, of course more literature is available. A selection is added to Table 1.

Page 6, Page 6, line 201: I don't follow this sentence. Do you mean SOA is being formed from the oxidation of first generation products rather than the products themselves?

The strong contribution of first-generation oxidation products to SOA formation was developed and reported in detail by Ng et al., 2006. A delay time in growth curves often indicates that further reaction of oxidation products is able to yield condensable products. The sentence as changed.

**Page 6, line 205**: *Such a long induction period is most likely caused by further reaction of first-generation oxidation products leading to SOA formation as it was demonstrated in previous studies (Ng et al., 2006, Mutzel et al., 2016).*

Page 6, line 204: Please give structures for the non-standard named compounds like Myrtenal. Myrtenal is not a non-standard named compound. Myrtenal is the trivial name of (1*R*)-6,6-Dimethylbicyclo[3.1.1]hept-2-en-2-carboxaldehyde. It has a CAS number 18486-69-6 and is commercially available.

Page 6, line 216 –please give typical limonene ozonolysis SOA yields.

Yield from ozonolysis are added to the manuscript.

**Page 6, line 217:** *Those values are close to the lowest values reported for limonene ozonolysis with 29 – 69 % (Northcross and Jang, 2007, Chen and Hopke, 2010, Gong et al., 2018).*

Page 7, line 222 –is there no literature data for SOA from cresol for comparison? The start of this section is rather imprecise. Using cresol as a proxy for "anthropogenic" VOCs and comparing whether they form more SOA than biogenics is not accurate. Hildebrandt uses the parent hydrocarbons and so cannot simply be applied to the cresol.

The respective sentence was deleted from the manuscript

**Page 7, line 222** was deleted: *A study by Hildebrandt et al., 2009 raised the question about the low SOA yields and observed much higher yields by using artificial sunlight.*

Cresol is an anthropogenic VOC as well as a biomass burning tracer compound. The authers don´t think that this is contradicting. Biomass burning can be of anthropogenic origin. Available literature for cresol is cited many times throughout the manuscript.

A comparison of the SOA yields is done because biogenic VOCs are emitted during day-time. During evening biomass burning starts and compounds like cresol are emitted. This mixture of VOCs and OVOCs is subject to oxidation by OH and $NO_3$. This was exactly the objective of the present manuscript and is discussed in detail in section 3.5 Atmospheric implication.

Page 7, line 240 –there is no mention here about the impact of RH on aerosol phase and hence the uptake of SVOC.

Indeed, this investigation of SVOCs and their partitioning in the particle phase might be relevant and could be objective of a follow-up study. But it was no focus of the present study.

Page 7, line 250: You have not mentioned limonene + OH, which seems to have a different pattern to the others, with the highest yield at 50 % RH?

As it stated in the manuscript "According to **Fehler! Verweisquelle konnte nicht gefunden werden.**, a significant effect can be observed for two systems, i.e. limonene/$NO_3$ and *m*-cresol/OH while α-pinene/$NO_3$ and *m*-cresol/$NO_3$ were not affected by RH in good agreement with the literature studies by Bonn and Moortgat 2002, Fry et al., 2009, Boyd et al., 2015)" these two cases were discussed in detail.

Page 8, line 277: can you really say this based on a SAR derived rate constant from the MCM?

As no experimental data are available, no other values can be provided. It clearly written, that this is a hypothesis. The SAR might be not completely accurate but the difference is at least two order of magnitude.

Page 8, line 280: Were these two sets of reactions done in this study? What is the "one study that investigated this reaction channel"?

This paragraph summarizes the study by Boyd et al., 2015. He investigated the reaction of $RO_2+NO_3$. To the authors best knowledge this hasn´t subject of other studies.

Page 8, line 297 –I don't understand what "the evolution in dependency on consumption" means.

This refers to the signal measured at *m/z* 123 which is illustrated as a function of the consumption. The sentence was changed accordingly.

**Page 8, line 297**: *Nevertheless, the increase of the signal in dependency on consumption does not show a significant effect of RH on the formation (Fehler! Verweisquelle konnte nicht gefunden werden.).*

Page 9, line 305: This whole section is really confusing, with long sentences with too many parts. There needs to be some more discussion about the data in the figure. Why are some values over 100 %? The proportion of OM also is much lower in limonene + NO3, why is this? I felt there was a lot of information in the figure, but very little discussion of what it means and the levels of uncertainty.

The authors do not agree on this comment. The discussion at page 9 "*Organic carbon and water-soluble organic carbon*" summarized the findings according to Figure 4. But in Figure 4 only sum parameters are provided. The break-down those sum parameters into smaller fractions, single discussions are provided regarding the peroxide content and single compound analysis. The following sentence is added to the manuscript to make it more clear:

**Page 9, line 326**: *To further investigate the fraction of organic material found in the formed SOA, discussions about SOA bound peroxides and single compounds is provided in the following sections.*

The level of uncertainty is given for each used technique (Figure 4). These uncertainties are the reason for values being slightly larger than 100%. Figure 3 illustrates the combination of three independent measurements, one online and two offline techniques. Small deviations are expected.

Also a discussion on the small fraction of water-soluble material in the case of limonene is provided. Although, the present work couldn´t find the reason for such a low fraction, feasible reasons are provided.

Page 9, line 321: What is the sensitivity of this analysis as a result of changing the molar mass? There is increasing evidence of auto-oxidation in some systems, which can occur in the presence of NO, which could shift the peroxide mass distribution.

The peroxide test gives an impression about the amount of hydroperoxides in the SOA. Acceding to Eq. 2-4 it can be seen that this analysis is very sensitive towards the molar mass. A molar mass of 300 g mol$^{-1}$ is usually used as it refers to the original publication by Docherty et al., 2005 assuming that the majority of hydroperoxides are higher molecular weight compounds. This fact is clearly stated in the manuscript. While changing the molar mass, the fraction will change. As long as the identification of this fraction cannot be provided on a molecular level, it is valid to use those assumptions.

Page 10, line 355 to 360: can you make it clear that you are talking about HO2 concentrations in the GAS phase.

This information was added to the paragraph.

**Page 10, line 360:** *Anyway, the observed trend of lower peroxide fractions under elevated RH is consistent in both studies and might be a result of two facts, i) the uptake of HO$_2$ radicals from the gas phase and ii) decomposition and/or hydrolysis of hydroperoxides. It has been reported that the gas phase HO$_2$ radical concentration is significantly suppressed by three orders of magnitude when a liquid phase is present (Herrmann et al., 1999). Thus, the HO$_2$ uptake might increase under elevated RH and therefore HO$_2$ in the gas phase is only available to a lesser amount to react with RO$_2$ radicals (Herrmann et al., 1999). Furthermore, under high RH decomposition and/or hydrolysis occur to a larger extent lowering the peroxide fraction (Chen et al., 2011, Wang et al., 2011).*

Page 11, line 415. I don't follow this sentence. To me the RH dependence looks very different –nowhere near as dramatic at 75 %

The sentence was changed accordingly.

**Page 11, line 423:** *Notably, pinonic acid was detected in comparable amounts from a-pinene/OH and a-pinene/NO$_3$ with the same RH dependency (**Fehler! Verweisquelle konnte nicht gefunden werden.***).*

---

## Author Comment (AC2) · 12 Oct 2020

We gratefully thank the reviewer for the comments. Please find the corresponding answers below. Changes in the manuscript are highlighted in blue.

**Reviewer 2**

Please use abbreviation consistently. It improves the readability of the manuscript greatly. E.g. RH and relative humidity.

L23 In average -> On average

The manuscript was changed accordingly and now reads:

**Page 1, line 23:** *On average, α-pinene yielded SOA with about 6 - 7% from $NO_3$ radicals and $3 – 4$ % from OH radical reaction.*

L127: Please give the name of an IC-CD model and manufacturer.

Thermo Scientific Dionex ICS-3000 CD

L150: Please provide a reason for changing the chromatographic condition.

The chromatographic separation was done accordingly to Hoffmann et al., 2007. These authors figured out that these conditions are much better for separation. To emphasize that the method is taken from Hoffmann and co-workers (from our lab), the citation is added.

**Page 5, line 149:** *For anthropogenic SOA compounds, the separation was done as described above at 15°C and with 0.2 % acetic acid in water as described in Hoffmann et al., 2007.*

L186: Do the authors mean "The values obtained in this study did not agree with previously reported values because OH sources are different"?

Yes, this is correct. As the choice of OH-radical generation is crucial for product distribution (eg. Influence of $RO_2/HO_2$ ratio etc.), the OH source might have an effect on $\alpha_{1/2}$ and $K_{1/2}$ values.

L239: This should be 3.2. Please change all the section numbers accordingly.

The section numbers were changed accordingly:

**Page 7, line 243**: 3.2 *Influence of RH on SOA yield and growth*

**Page 9, line 308:** 3.3 *Characterisation of particle-phase chemical composition*

**Page 13, line 456**: *3.4 Atmospheric implications and Conclusion*

**Page 14, line 501**: *4 Acknowledgment*

**Page 14, line 508:** *5 References*

Line 262: I am not too sure what the authors mean here by "a decreasing consumption when RH increases". Do the authors mean "a decreasing consumption of VOC when RH increases"? If so, this seems to contradict what Figures 1 and 2 (limonene/NO3) shows. Please clarify this sentence.

This description is not contradicting. As it can be seen in Figure 1, in the case of limonene/NO3 the consumption of limonene is lower under dry conditions ($\Delta HC = 193$ µg m$^{-3}$) than under humid conditions ($\Delta HC = 107$ µg m$^{-3}$). Thus VOC consumption decreases when RH is higher. Additional information is added to make it more clear. The text now reads:

**Page 8, line 269:** *As can be seen in Figure 1, in the case of limonene/NO3 the consumption of limonene is lower under dry conditions ($\Delta HC = 193$ µg m$^{-3}$) than under humid conditions ($\Delta HC = 107$ µg m$^{-3}$).*

L307: NPOM should be described in line 73.

**Page 9, line 316:** *Please note, water-soluble organic carbon was determined as NPOM.*

**Page 1, line 72:** *The chemical composition of formed SOA was characterized for their fraction of organic carbon (OC), non-purgeable organic carbon (NPOM), SOA-bound peroxides and SOA marker compounds.*

L315: It may well be that higher molecular weight compounds do not ionize well in negative ESI, and they aren't present in the LC/MS data. Just in case, do the authors have signals for them in LC/MS data? Signals were present but only few and difficult to interpret. As the ionization efficiency is unknown and matrix effects might play a role, an interpretation of those data were not done.

L346: a-pinene/OH -> α-pinene/OH
The manuscript was changed accordingly and now reads:
**Page 10, line 358:** *In general, peroxide fractions of 10 - 80% of the organic mass have been detected from α-pinene/OH experiments.*

L346-347: Why do high peroxide fractions contradict the small SOA yields? Please provide the reason for this explanation.
The sentence is deleted from the manuscript.
**Page 10, line 358:** *The high peroxide fractions of 10 – 80% are contradicting to the small SOA yields from α-pinene/OH ($Y_{OH} \approx 3.5$ %)* is deleted

L376 Biogenic SOA marker compounds and Figure 6: Does the discussion take the water content of SOA? If not, it makes more sense to discuss data in terms of carbon mass fraction of marker compounds in OC to eliminate the effect of water content.
The effect of LWC on the partitioning behavior and formation processes is discussed in detail at page 11 and 12. As pointed out by the reviewer, the amount of marker compounds is expressed as mass fraction in formed OM. Thus, dilution effects are eliminated.

L865 Figure 4 Caption: non-purgeable organic material (OM) -> non-purgeable organic material (NPOM).
The manuscript was changed accordingly.
**Page 26, line 885:** *Comparison of organic mass calculated from SMPS with an offline determined concentration of organic material (OM) and non-purgeable organic material (NPOM).*

L415: Do the authors mean pinonic acid was detected in comparable fractions in both NO3 and OH oxidation of α-pinene? If so where do these 20-25% come from? For the 75%RH experiments, α-pinene/OH experiment shows much lower pinonic acid fraction. Can the authors clarify this?

Pinonaldehyde was detected from both systems and the amounts were comparable. To clarify, the sentence was re-written and now reads:
**Page 12, line 427:** *Notably, pinonic acid was detected in comparable amounts from α-pinene/OH and α-pinene/NO₃ with the same RH dependency (Figure 6).*

At this moment, it is unclear why pinonic acid is formed less at RH = 75%. As this was not observed for α-pinene/NO₃ system, water might affect the formation process of pinonic acid during OH-radical reaction. This fact remains still speculative and more work is needed to evaluate this observation.

---

## Referee Report (RR1)

**Review of Muntzel et al. 2020 "Importance of SOA formation of a-pinene, limonene and m-cresol comparing day- and night-time radical chemistry"**

**Summary**

The authors present a study about the SOA formation potential of three precursors that are relevant for the ambient atmosphere. They compare the SOA mass yield and SOA particle composition for two different formation pathways – reaction with OH radicals (representing daytime photochemistry) and reaction with NO3 radicals (nighttime chemistry) – while also investigating the impact of different humidity regimes. As there is limited data available especially for the NO3 radical chemistry, this is an important investigation which will help to improve the understanding of SOA formation processes from a range of different precursors and oxidants.

However, certain issues have to be address prior to publication in ACP. The main problem lies with the still unclear presentation of the work. Several pieces of basic information are either not stated clearly or left to be guessed by the reader (see below for examples).

**Major Comments**

1) There is no information about the VOC and NOx concentrations. Concentrations (or even better exposure values) for NO3 and OH radicals are also missing. This makes it very difficult to relate this work to the existing literature. Even if these values may be stated in a previous paper about this data set, it is not acceptable to ask the reader to search for this vital information. At least the starting VOC and average NOx concentration must be stated (e.g. added to Table

1). At least an estimate of the OH and NO3 exposure must be given. What does 90 min of reaction time in the chamber represent? Is it equivalent to an average day/night of reactions at ambient conditions? Or is the exposure much greater.

2) The authors do not explain how the yield values were calculated. It may seem trivial to them, but it is important to be stated. Also, it has to be clarified that each yield curve in Fig 1 stems from a single chamber experiment and thus represents the evolution in reaction time as well as in formed organic mass space.

3) One key finding of this study is the difference in SOA mass yield. It is commendable that the authors present the yield vs organic mass curves (Fig 1) and then use the approach from Odum et al. to take their shape into account. But when the overall yields are compared between the precursors, the authors do not clearly state that now they are comparing the yields at the end of their 90 min chamber experiment. By doing this, they completely ignore the different turnover in their experiments (i.e. how much of the VOC reacted). The overall interpretation should not change. But if the experiments are compared for the same ΔM value, the detailed interpretation may differ. As an example, for the 75%

RH case, cresol+OH has a higher yield at $\Delta M=10$ ug m$^{-3}$ than limponene+OH. Guessing that the same initial VOC concentration was used in both experiments, this behaviour should yield information about differences in reaction rates for the two precursors. This is of interest as there is so limited data about these reactions. For future studies, it may be advisable to also conduct experiments with higher initial VOC concentration to compensate for a slower reaction rate. For the manuscript I request that the authors clearly state that they compare the yield ($t_{reaction}=90$min).

4) The authors compare their yields after 90 min reaction time of an unknown amount of VOC with an unknown amount of OH or NO3 radicals to the yield values measured in other studies. As they clearly show in their Fig 1, the SOA mass yields are highly dependent on the reaction time or the formed organic mass. How comparable are all these values in the literature to your experiment settings? Did you only compare literature values with comparable $\Delta M$? Or did you base the comparability on the reaction time? Or the VOC turnover ($\Delta VOC$)? Or the exposure to OH/NO3? Please clarify how you selected the studies and if necessary adjust the examples selected in the manuscript.

5) Similar to a previous reviewer, I wonder why the authors call cresol SOA "anthropogenic" if its main source is biomass burning which is not generally considered "anthropogenic". I am not satisfied with the reply to the previous reviewers comment. Just because cresol is an aromatic compound does not automatically mean it stems from an anthropogenic source as is typically assumed for benzene or toluene. Additionally, the connection of cresol to biomass burning is not mentioned in the introduction but only at the end of section 3.3. The authors need to shift this information to the introduction and clearly state that while a-pinene and limonene are biogenic VOC, cresol is a VOC mostly connected to biomass burning. The term "anthropogenic" should be replaced by something more fitting (e.g. aromatic).

6) The SOA-bound peroxides value for the dry a-pinene+OH experiments seems extremely high and such a value would have important implications for the interpretation of many measurements. The authors speculation for the reasons for that seem plausible. However, I assume the authors did not conduct any repetitions of the chamber experiments (as it is very time consuming). Your interpretation is then based on a single filter sample from a single experiment. How reliable is that? I.e., is it possible that "something went wrong" in collecting and handling the filter or during the peroxide measurement?

7) The overall language needs polishing in a lot of places and would benefit from the input of a native English speaker. The structure of many sentences is closer to German than to English which makes the manuscript difficult to follow. Overall, the language style is inconsistent and gives the manuscript a "sloppy" character (see specific comments for examples). This has been noted by previous reviewers and – in my opinion – has not improved since the previous manuscript version.

**Specific comments**

+ In several places, acronyms/abbreviations are not properly introduced (e.g. BSOA in line 66, QF filter in line 92, or HMWC

in line 422 to name just a few). The authors must introduce all acronyms/abbreviations and then use the introduced terms in the following.

+ The use of "day-time", "day time" and "daytime" is not consistent within the manuscript. According to the online Cambridge dictionary and Merriam-Webster the noun should be "daytime" (and nighttime, respectively). ACP may have a different preference, but in any case, chose one spelling and stick with it.

+ The term "the present study" should be replaced with "this study" as it is much more reader friendly. In the context of this manuscript, it is always clear that the authors are referring to their own work. However, the ACP style guide may differ from my opinion.

+ The term "under RH" is used several times. It should be "at elevated RH" or "under higher RH condition".

+ The authors switch between VOC+NO3 and VOC/NO3 for their experiment labels. Decide which you like and stick with it.

+ L 22ff: The statement that the SOA formation potential of limonene with NO3 significantly exceeds that of the reaction with

OH is only strictly true for the dry case and when comparing the $t_{reaction}$=90min points. For 50% and 75% RH this is not correct.

The yields are comparable or even higher for OH. The authors need to correct this statement.

+ L 44: BVOCS acronym not introduced

+ L 51ff: this sentence is difficult to understand. Rephrase to make clear what is causing what.

+ L 53ff: The structure of the sentence is very difficult to follow. (paraphrasing: "The reaction is suggested to be more important for BVOCs than AVOC being the reason…" rephrase.

+ L53ff: Here is the first time the authors bring up biogenic vs anthropogenic VOC when discussing the different chemical reaction behaviour. In this respect, the main important functionality in the compounds is the aromaticity of many anthropogenic

VOCs. Considering cresol being a biomass burning tracer and not necessarily a true anthropogenic VOC, "anthropogenic"

should be changed to "aromatic" to highlight that this is the real reason for the different chemical behaviour. (Also adjust the label in L 72.)

+ The chamber description in section 2.1 is missing some important details. Yes the previous paper contains these details, but the following basic information needs to be directly available to the reader of this paper:

-    It is not clearly stated if the chamber is operated in "batch mode" or "flow through". Is the removed sample volume replenished or does the chamber change size?

-    What is the material of the chamber?

-    Were the particles dried prior to injection? Are they in liquid or solid phase state when equilibrating to the RH

inside the chamber?

+ L 82: Is the chamber itself actively humidified or does the humidifier condition the air that is used to flush/fill it?

+ L 88: You speak of a pre-reactor, then of a flow tube. It is not clear that they are the same thing.

+ L 91: "samples were taken at filter" ? Please correct this to whatever you mean.

+ L 92: "QF filter" neither is the acronym explained nor the brand/make stated.

+ L 100: PTR-TOFMS acronym not introduced.

+ L 101: according to what?

+ L 103: You used an intermediate RH to determine particle wall losses. How representative is that for the dry experiment? dry and wet deposition may be different. Especially if the phase state of the particles changed.

+ L 106: add a comma between reaction and conditions to clarify the sentence meaning.

+ L 117: The authors replied to the previous reviewer question that they use $1.0$ g cm$^{-3}$ as particle density because the real density of the particles is not known. The way this is phrased here suggests that $1.0$ g m$^{-3}$ is the true density (2 out of 3 readers misunderstood this information). It needs to be added in the text that in absence of any reliable density estimation, $1.0$ g m$^{-3}$ is used instead.

+ L 118: was the water content accounted for when calculating the change in organic mass $\Delta M$? I.e., was the whole volume as measured by the SMPS considered to be organic material? Or was a fraction subtracted according to the hygroscopic growth estimated for SOA at 50% and 75%?

+ L 136: why is "biogenic and anthropogenic marker compounds" an italic subheading instead of adding it to the section heading?

+ L 130: IC-CD acronym not introduced

+ L 144: HPLC acronym not introduced

+ L 156: add the purity information for the pure chemicals from commercial sources.

+ How was the yield calculated. What assumptions were made?

+ L 177: Eq. (Eq. 1) correct typo

+ L 178: it should be $M_0$ both times. not M

+ L179: $M_0$ is not listed

+ L 180: add "the" before mass yield

+ L 185: tiny tables in Fig 1. Even at 200% zoom, the numbers in the tables in Fig 1 are barely readable. Increase font size.

+ L 189: Can you really directly compare the parameters from a one product Odum fit with a two product fit? Have you tried fitting your data with a two product fit? How do the parameters compare?

+ L 200: It is not clear which of the 4 Fry et al. references is meant here.

+ L 212: "…proceeds via further reaction… " this is an ambiguous statement. Rephrase to clarify.

+ L 213: Isn't the limiting factor that the first-generation products are not of sufficiently low volatility to partition in the particle phase at the organic mass loading at that time? Further oxidation creates compounds of lower volatility which will contribute to particle mass. A different reaction pathway forms products with a different volatility distribution.

+ L 216: In the presence of NOx, organonitrates should also form in the OH experiments.

+ L 219 yield from limonene + NO3

+ L 220ff: 16-29% is close to 29-69%? I disagree. I also disagree with your statement that the NO3 reaction yields more SOA than ozonolysis.

+ L 224 ff: You cannot compare the importance of NO3 vs OH vs O3 without stating how your experiment conditions relate to ambient conditions (what is the equivalent OH/NO3 exposure or oxidative age?). Unrealistically high NO3 values would distort your findings. AS you do not give these values one can only wonder.

+ L 230ff: how much SOA mass was formed in your blank experiments (seed, oxidants but no VOC)? How does this "chamber background" SOA compare to the SOA mass formed in cresol+NO3?

+ L 241: "…was used." ? was used to form OH?

+ L 249: "Aa discussed before…" N2O5 has not been mentioned anywhere before this point in this manuscript. Adjust the wording to reflect that.

+ L 253: "…- as to the authors' knowledge…" remove the "as"

+ L 262: remove "Contrary". This implies a connection to the previous sentence that is not there.

+ L 274:"Fig 1"? should that not be Fig 2 here?

+ L 274f: The numbers in parenthesis do not match the sentence meaning. 60 is larger than 20 but the statement is opposite.

+ L 280: change "suggested" to "assumed" or "inferred"

+ L 281ff: Why would the behaviour of endolim create a RH dependent response? nothing in your explanation suggests a RH dependency.

+ L 291ff: same as for the previous paragraph. Why does this behaviour explain the observed RH dependence?

+ Regarding the RH dependence of Cresol+NO3 SOA yields. Have you considered a process analogue to the formation of isoprene SOA? There, smaller, semi-volatile but water-soluble compounds are taken up into the particle phase where they oligomerise and for low-volatility products. An aqueous phase enhances this by drawing the water-soluble compounds from the gas phase and also catalysing the condensed-phase reactions especially if $SO_4^{2-}$ ions are present (forming low volatility organosulfates).

+ L 310: Fig 7 is mentioned before figures 3-6.

+ Fig 7: MNC is -5-nitro in the caption but -6-nitro in the figure label and the main text.

+ L 315: OC and WSOC are introduced here after they have already been used earlier. Ensure that acronyms/abbreviations are written out when they first occur and use the acronym/abbreviation consistently throughout the rest of the text.

+ L 319: What is the relationship between WSOC and NPOM. It reads here (and in the methods section) as if these two names are interchangeable. Is that the case? Is WSCO derived from the NPOM measurement?

+ L 319: The authors speak of OC/EC and WSOC. But Fig 4 shows OM and NPOM. It is not explained how OM relates to

OC/EC. It may seem trivial to the authors. But you cannot expect the general ACP reader to know all your measurement equipment and what quantities are measured/derived.

+ L 320ff: When comparing the SMPS derived mass values with other directly mass based measurements, it is very important that the authors state their assumptions for the density, i.e., that it is set to 1.0 g cm$^{-3}$ and is constant for all conditions. In comparable chamber experiments particle densities of 1.2 – 1.4 g cm$^{-3}$ are typically derived. This means that the reported $\Delta M(SMPS)$ are most likely 20 – 40% too low. This has to be at least mentioned and considered when giving the uncertainties for the SMPS (caption of Fig 4). I recommend that the authors focus more on the qualitative trends $\Delta M(SMPS)$ and OM (and NPOM, respectfully). This agreement may be an indication that the density does not change significantly with RH for these experiments.

+ Fig 4: it is not directly clear what the % values refer to. Add this information to the Figure caption.

+ Fig 4 the uncertainty regarding the particle density needs to be mentioned when stating the uncertainty for the SMPS measurement. From the current wording it is not clear if the 10% already are supposed to take the density uncertainty into account.

+ L 322: "..except mass originated from limonene" replace with "…except in the limonene experiments"

+ L 324: replace "compound" with "precursor"

+ L 325: remove"is" (… with x% of org mass composed of…)

+ L 325ff: Have the authors considered that in the limonene+NO3 case a large fraction of hydrophobic products may be produced. An aqueous phase could then hinder the uptake of these compounds (quasi reverse co-condensation) which then stay in the gas phase leading to a lower SOA mass yield. It does seem odd that this happens in the NO3 case where the authors expect a higher organonitrate content. In general, the polar nitrate groups should make the molecules more hydrophilic

+ L 326: Now the term "WSOM" is used. How does that relate to NPOM and WSOC? These inconsistencies make the manuscript hard to follow.

+ L 325ff: At least the name of the measurement principle for the peroxide measurement should be given here or better in the methods sections.

+ L 336: change "it" to "is"

+ L 337: Is the "mostly dimer" assumption still valid considering the more recent information about the highly oxygenated material (HOM, (Ehn et al., 2014)) that is produced in auto-oxidation processes. Most of these compounds contain hydroperoxide groups but are not necessarily dimers.

+ The absence of a RH dependence my be connected to the higher content of water-insoluble material in limonene SOA. If enough water insoluble material is present, a separate organic phase may form in which the peroxide compounds are "protected" from hydrolysis.

+ L 338: What is "the organic mass formed during the experiment" referring to ($M_{org}$)? The mass as derived from the SMPS measurement? Then the same thoughts regarding the uncertainty of the particle density apply here and should be clearly stated.

+ L 365ff: It is impossible to evaluate this comparison as no NOx values are given for the experiments in this study. Terms like low/medium/high are very subjective.

+ regarding possible artefacts in the peroxide measurements: How much H2O2 is left in the chamber? H2O2 uptake into particles may play an important role if very high concentrations of H2O2 were present (e.g. ppm level). The statement "blank filter were carefully checked" is not clear. Are these the filters collected after the blank experiments? Or are these blank filters? Will the uptake of H2O2 be the same on pure (NH4)2SO4/H2SO4 particles as on mixed inoganic/organic?

+ L 393ff: It is not clear to me what the marker compounds are normalised to. The SMPS derived organic mass? Or the OM?
Or the HPLC Signal?

+ L 406f: LWC and ALWC are not introduced. Is there a difference between the two quantities? Or is this another one of the inconsistencies that are spread throughout this manuscript?

+ L 422: HMWCs is not introduced.

+ L 450: state also the higher values to make the comparison with the low RH values easier.

+ L 466: The O in ROS stands for oxygen – not organic! This term refers to the species creating oxidative stress in the system. These compounds can be organic peroxides. But it is definitely not referring to all organic. Non peroxy organic aerosol also have adverse health effects but via different mechanisms.

+ L 482: The structure of the sentence "The formation of pinonic acid…" does not make sense in English. "proceed" is not the correct term when talking about chemical formation mechanisms. A phrase such as "Pinonic acid might be formed from
the oxidation of pinonaldehyde that itself has been found for NO3-radical initiated reactions of a-pinene." is much clearer and describes the ongoing processes more precisely.

+ L 491 "pointed out to be" do the authors mean "turned out to be"?

+ L 495: The reactive species in the gas phase are only a reservoir for particulate matter if the reaction products have a sufficiently low vapour pressure or if other mechanisms form low volatility materiel (e.g. particvle phase oligomerisation as
in the isoprene SOA case).

---

## Author Response (AR2)

We thank the reviewer for the comments which, we think, led to further improve the manuscript. Please find the corresponding answers below. Changes in the manuscript are highlighted in blue.

*1) There is no information about the VOC and NOx concentrations. Concentrations (or even better exposure values) for NO3 and OH radicals are also missing. This makes it very difficult to relate this work to the existing literature. Even if these values may be stated in a previous paper about this data set, it is not acceptable to ask the reader to search for this vital information. At least the starting VOC and average NOx concentration must be stated (e.g. added to Table 1). At least an estimate of the OH and NO3 exposure must be given. What does 90 min of reaction time in the chamber represent? Is it equivalent to an average day/night of reactions at ambient conditions? Or is the exposure much greater.*

Authors response:

We thank the reviewer for his comments. We have tried to make the experimental conditions more clear in our present revision. The concentration of OH and $NO_3$-radicals have now been included into the manuscript. The OH radical concentration was determined by the OH-clock determination described by Bartmet et. al., 2011. $NO_3$ radical reaction was estimated by implementing the kinetic box model described by Fry et al., 2014 in the complex pathway simulator (COPASI).

**Page 3, line 89:** '*Applying the method developed by Barmet et al., 2011, the average OH radical mixing ratio in the chamber is about $3 – 5 \times 10^6$ molecules cm$^{-3}$.* '

**Page 3, line 93:** '*Including the kinetic box model developed by Fry et al., 2014 into the COPASI (COMPLEX PATHWAY SIMULATOR), the mixing ratio of $NO_3$ radicals is calculated for the present study to be $7.5 \times 10^7$ molecules cm$^{-3}$. The reaction mechanism provided by Fry and co-workers we implemented into COPASI and the model was utilized to the aerosol chamber.* '

The initial hydrocarbon concentration was already given in caption of Table 1. To highlight this essential information it was now also added into the main chamber description at MS page 3. The same was done for the initial NO concentration.

**Page 3, line 85:** '*All experiments were done with an initial hydrocarbon mixing ratio of 60 ppbv.* '

**Page 23, line 89:** '*OH-radical experiments were done in the presence of 10 ppbv NO.* '

In addition data from experiment at RH=50% ($\alpha$-pinene/OH, $\alpha$−pinene/NO$_3$, limonene/OH, limonene/NO$_3$, cresol/NO$_3$) are available at the EUROCHAMP webpage (https://data.eurochamp.org/).

Even that the concentration of OH and $NO_3$-radicals are much higher than under ambient conditions, the conducted experiments provide important insights into the $NO_3$ and OH radical chemistry of a-pinene, limonene and cresol. For future studies experiments under continuous flow conditions should be considered because those experiments

*2) The authors do not explain how the yield values were calculated. It may seem trivial to them, but it is important to be stated. Also, it has to be clarified that each yield curve in Fig 1 stems from a single chamber experiment and thus represents the evolution in reaction time as well as in formed organic mass space.*

Authors response:

To better clarify, additional information is now given in main manuscript and in the figure caption. This text reads as follows:

**Page5, line 173:** '*The SOA yields were calculated according to Odum et al., by calculating the amount or produced organic mass in relation to the amount of reacted hydrocarbon.*

$$Y_{SOA} = \frac{\Delta M}{\Delta HC} \qquad\qquad (Eq.\ 1)$$

*where*

*$\Delta M$ is the produced organic mass [µg m$^{-3}$]*

*$\Delta HC$ is the reacted amount of hydrocarbon [µg m$^{-3}$]*

**Page 24, line 885:** *Each fit present a single chamber experiment. SOA yield was calculated by deviating the produced organic mass by the consumed amount of hydrocarbon.* '

*3) One key finding of this study is the difference in SOA mass yield. It is commendable that the authors present the yield vs organic mass curves (Fig 1) and then use the approach from Odum et al. to take their shape into account. But when the overall yields are compared between the precursors, the authors do not clearly state that now they are comparing the yields at the end of their 90 min chamber experiment. By doing this, they completely ignore the different turnover in their experiments (i.e. how much of the VOC reacted). The overall interpretation should not change. But if the experiments are compared for the same $\Delta M$ value, the detailed interpretation may differ. As an example, for the 75% RH case, cresol+OH has a higher yield at $\Delta M$=10 ug m-3 than limponene+OH. Guessing that the same initial VOC concentration was used in both experiments, this behaviour should yield information about differences in reaction rates for the two precursors. This is of interest as there is so limited data about these reactions. For future studies, it may be advisable to also conduct experiments with higher initial VOC concentration to compensate for a slower reaction rate. For the manuscript I request that the authors clearly state that they compare the yield (treaction=90min).*

Authors response:

To avoid any misunderstanding, we followed the suggestion by the reviewer and included a respective sentence at Page 5, that the discussion of the SOA yield is mainly focused on the final SOA mass produced after 90 minutes. It is also mentioned that a comparison of the curve shape of the growth curves is done. The sentence reads:

**Page 5, line 184:** *'Throughout this study, it is mainly referred to the amount of SOA mass produced measured after 90 minutes reaction time. Only the differences in curve shape of growth curve are discussed in detail in the respective section.'*

*4) The authors compare their yields after 90 min reaction time of an unknown amount of VOC with an unknown amount of OH or NO3 radicals to the yield values measured in other studies. As they clearly show in their Fig 1, the SOA mass yields are highly dependent on the reaction time or the formed organic mass. How comparable are all these values in the literature to your experiment settings? Did you only compare literature values with comparable $\Delta M$? Or did you base the comparability on the reaction time? Or the VOC turnover ($\Delta VOC$)? Or the exposure to OH/NO3?*

Authors response:

In Table 1 a complete overview about $\Delta HC$ for each single experiment is given. The comparison was done for experiments with comparable experimental conditions. There is only a limited number of

studies providing exposure levels for chamber experiments. Therefore, the comparison was focused on comparable experimental conditions. Studies from oxidation flow reactors (OFR) were not included.

*Please clarify how you selected the studies and if necessary adjust the examples selected in the manuscript.*
Authors response:

Please see comment above. This selection is now clarified by the footnotes to Table 1 which now reads:

**Page 23, line 892:** *"only those studies are reported for OH radical reaction of limonene and $\alpha$–pinene that apply also $H_2O_2$/NO as OH source; [b] due to the lack of data all available literature is shown'*

*5) Similar to a previous reviewer, I wonder why the authors call cresol SOA "anthropogenic" if its main source is biomass burning which is not generally considered "anthropogenic". I am not satisfied with the reply to the previous reviewers comment. Just because cresol is an aromatic compound does not automatically mean it stems from an anthropogenic source as is typically assumed for benzene or toluene. Additionally, the connection of cresol to biomass burning is not mentioned in the introduction but only at the end of section 3.3. The authors need to shift this information to the introduction and clearly state that while a-pinene and limonene are biogenic VOC, cresol is a VOC mostly connected to biomass burning. The term "anthropogenic" should be replaced by something more fitting (e.g. aromatic).*

Authors response:

The term anthropogenic is not longer used. It is substituted by aromatic. The connection to biomass burning is also shifted towards the introduction. This text now reads: '

**Page 2, line 68:** '*This study is aimed to investigate three selected precursor compounds, namely $\alpha$-pinene and limonene as biogenic VOCs and m-cresol as aromatic VOC with regards to their SOA formation potential under nighttime ($NO_3$ radicals) and daytime conditions (OH radicals). While a-pinene and limonene are important BVOCs, m-cresol is often related to biomass burning.'*

*6) The SOA-bound peroxides value for the dry a-pinene+OH experiments seems extremely high and such a value would have important implications for the interpretation of many measurements. The authors speculation for the reasons for that seem plausible. However, I assume the authors did not conduct any repetitions of the chamber experiments (as it is very time consuming). Your interpretation is then based on a single filter sample from a single experiment. How reliable is that? I.e., is it possible that "something went wrong" in collecting and handling the filter or during the peroxide measurement?*

Authors response:

We agree to this concern of the reviewer. Aside from the fact that experiments can go wrong, one fact is of higher importance. SOA-bound peroxides are determined by iodometric detection. As described in Docherty et al., 2005, the fraction of peroxide is afterwards calculated assuming a molar mass of 300 g mol-1. This only valid if higher molecular weight compounds are present. Assuming that those peroxides might be also of another nature (methylhydroperoxide), the molar mass can differ a lot and thus, the total fraction. Consequently, the assumption of the molar mass introduces the highest uncertainty for the calculating the fraction of SOA-bound peroxides. To address this issue an additional sentence is added, which now reads.

**Page 10, line 350:** '*Organic peroxides in SOA were quantified according to a method published by our laboratory (Mutzel et al., 2013), assuming a molar mass of 300 g mol$^{-1}$ (Figure 5), as is recommended by Docherty et al., 2005, presuming that the majority of organic peroxides are higher-molecular weight compounds (e.g. dimers). Notable, the assumed molar mass has a significant influence of the calculated amount of SOA-bound peroxides. This might cause some uncertainties. The method applies an iodometric detection by UV/Vis spectroscopy.*'

*7) The overall language needs polishing in a lot of places and would benefit from the input of a native English speaker. The structure of many sentences is closer to German than to English which makes the manuscript difficult to follow. Overall, the language style is inconsistent and gives the manuscript a "sloppy" character (see specific comments for examples). This has been noted by previous reviewers and – in my opinion – has not improved since the previous manuscript version.*
Authors response:

The authors agree on reviewers´ comment. The manuscript was revised by native speaker.

Specific comments

+ In several places, acronyms/abbreviations are not properly introduced (e.g. BSOA in line 66, QF filter in line 92, or HMWC in line 422) The authors must introduce all acronyms/abbreviations and then use the introduced terms in the following.

The manuscript was checked for abbreviations and all are now introduced upon their first use.

+ The use of "day-time", "day time" and "daytime" is not consistent within the manuscript. According to the online Cambridge dictionary and Merriam-Webster the noun should be "daytime" (and nighttime, respectively). ACP may have a different preference, but in any case, chose one spelling and stick with it.

The term 'daytime' and 'nighttime' is now used in an uniform manner throughout the complete manuscript.

+ The term "the present study" should be replaced with "this study" as it is much more reader friendly. In the context of this manuscript, it is always clear that the authors are referring to their own work. However, the ACP style guide may differ from my opinion.

The term was changed accordingly.

+ The term "under RH" is used several times. It should be "at elevated RH" or "under higher RH condition". +The authors switch between VOC+NO3 and VOC/NO3for their experiment labels. Decide which you like and stick with it.

'Under RH' was changed to 'At RH =' throughout the manuscript, as well as VOC+NO3 is no longer used.

+ L 22ff: The statement that the SOA formation potential of limonene with NO3 significantly exceeds that of the reaction with 75 OH is only strictly true for the dry case and when comparing the treaction=90min points. For 50% and 75% RH this is not correct. The yields are comparable or even higher for OH. The authors need to correct this statement.

The information that this statement is only valid under dry conditions is stated in the abstract. The abstract now reads:

**Page 1, line 22:** '*It was found that SOA formation potential of limonene with NO₃ under dry conditions significantly exceeds the one of the OH radical reaction, with SOA yields of 15 – 30 % and 10 – 21 %, respectively.*'

+ L 44: BVOCS acronym not introduced
The explanation is added. It now reads:

**Page 2, line 41:** '*The atmospheric degradation of biogenic volatile organic compounds (BVOCs) and subsequent SOA formation was subject of numerous studies during the last decades (Hallquist et al., 2009, Glasius and Goldstein 2016, Shrivastava et al., 2017).*'

+ L 53ff: The structure of the sentence is very difficult to follow. (paraphrasing: "The reaction is suggested to be more important for BVOCs than AVOC being the reason…" rephrase.

The sentence was re-written.

**Page 2, line 50:** '*The number of studies interconnecting NOₓ and BVOC emissions (Fry et al., 2009, Xu et al., 2015) are increasing, because the reaction with NO₃ is often considered to be more important for BVOCs than for AVOCs (Brown and Stutz, 2012).*'

+ L53ff: Here is the first time the authors bring up biogenic vs anthropogenic VOC when discussing the different chemical reaction behaviour. In this respect, the main important functionality in the compounds is the aromaticity of many anthropogenic VOCs. Considering cresol being a biomass burning tracer and not necessarily a true anthropogenic VOC, "anthropogenic" should be changed to "aromatic" to highlight that this is the real reason for the different chemical behaviour. (Also adjust the label in L 72.)

In the introduction "aromatic" is now used to highlight the different type of precursor compounds and the connection to anthropogenic SOA as well as biomass burning is included. The text now reads:

**Page 2, line 68:** '*While α-pinene and limonene are important BVOCs, m-cresol is often related to biomass burning. The chemical composition of formed SOA was characterized for their fraction of organic material (OM), water-soluble organic material (WSOM), SOA-bound peroxides and SOA marker compounds. For quantification of marker compounds, well known BSOA marker compounds (e.g. pinic acid, pinonic acid etc.) were used while SOA originated from m-cresol was characterized using a SOA mix than contains mostly anthropogenic SOA compounds that are often related to biomass burning (Hoffmann et al., 2007).*'

+ The chamber description in section 2.1 is missing some important details. Yes the previous paper contains these details, but the following basic information needs to be directly available to the reader of this paper:
- It is not clearly stated if the chamber is operated in "batch mode" or "flow through".

The chamber was operated under batch mode conditions. The corresponding information is added at Line 77 which now reads

**Page 2, line 77:** '*Experiments were conducted in the aerosol chamber under batch mode conditions at the Atmospheric Chemistry Department (ACD) of the Leibniz Institute for Tropospheric Research (TROPOS) in Leipzig.*'

Is the removed sample volume replenished or does the chamber change size?

The volume was not replaced and the chamber size changes. An additional information was added at line 93 which reads:

**Page 3, line 97:** *'During sampling time no additional air stream was added to chamber to avoid dilution.'*

- What is the material of the chamber?

The chamber is made of PTFE. The information is added at line 79.

**Page 3, line 84:** *'The aerosol chamber is made of PTFE and is of cylindrical geometry with a total volume of 19 $m^3$ and a surface to volume ratio of 2 $m^{-1}$.'*

- Were the particles dried prior to injection? Are they in liquid or solid phase state when equilibrating to the RH inside the chamber?

No dryer was used. After injection the particles were allowed to equilibrate with the chamber conditions. An additional information was added at line 84. It now reads:

**Page 3, line 84:** *'The seed particles were injected via a nebulizer without a dryer.'*

+ L 82: Is the chamber itself actively humidified or does the humidifier condition the air that is used to flush/fill it?

The chamber is humidified by passing a humid air stream through the chamber. An additional information is added.

**Page 3, line 87:** *'The humidifier is connected to the inlet air stream to enable humidification of air entering the chamber.'*

+ L 88: You speak of a pre-reactor, then of a flow tube. It is not clear that they are the same thing. Additional information was added and the text reads now as follows:

**Page 3, line 91:** *'$NO_3$ radicals were produced in a pre-reactor (operated as flow tube) by the reaction of $NO_2$ and $O_3$. A fraction of the air flow (10 L $min^{-1}$) out of the total air flow in the flow tube (30 L $min^{-1}$) was directed to the chamber.'*

+ L 91: "samples were taken at filter" ? Please correct this to whatever you mean.
The sentence was re-written and it now reads:

**Page 3, line 95:** *'After a reaction time of 90 min the reaction was stopped and samples were taken passing chamber air over a 47 mm PTFE filter (borosilicate glass fiber filter coated with fluorocarbon, 47 mm in diameter, PALLFLEX T60A20, PALL, NY, US) and QF filter (Micro-quartz fibre filter, 47 mm in diameter, MK 360, Munktell, Bärenstein, Germany), for 30 minutes at 30 L $min^{-1}$.'*

+ L 92: "QF filter" neither is the acronym explained nor the brand/make stated.
Acronym is explained. This now reads:
**Page 3, line 95:** *'After a reaction time of 90 min the reaction was stopped and samples were taken passing chamber air over a 47 mm PTFE filter (borosilicate glass fiber filter coated with fluorocarbon, 47 mm in diameter, PALLFLEX T60A20, PALL, NY, US) and QF filter (Micro-quartz fibre filter, 47 mm in diameter, MK 360, Munktell, Bärenstein, Germany), for 30 minutes at 30 L $min^{-1}$'*.

+ L 100: PTR-TOFMS acronym not introduced.
Explanation was added. It now reads:

**Page 3, line 107:** *'These values are within the measurement uncertainty of the Proton-Transfer-Reaction Time-of-flight Mass Spectrometer (PTR-TOFMS). '*

+ L 101: according to what?
Sentence was re-written. It now reads:

**Page 3, line 110:** *'According to the study by Romano and Hanna, 2018 an uncertainty of ±10% can be assumed and were thus not considered.'*

+ L 103: You used an intermediate RH to determine particle wall losses. How representative is that for the dry experiment? dry and wet deposition may be different. Especially if the phase state of the particles changed.
Yes, indeed the wall loss might change with RH. A RH = 50% was used as an approximation because it is an intermediate RH. A drastic effect of RH on particle wall loss is not expected because
- Experiments were conducted on a short time scale for 90 minutes. Previous studies demonstrated increasing wall losses with reaction time (McMurry 1985). It was found that for 90 minutes the wall loss is about 10% which is within the measurement uncertainty of the SMPS
- Although the phase state of the particle might change with the RH, this effect would be more pronounced if a dryer is used during particle injection. The particles are nebulized and injected into the chamber. Therefore they can regarded as wet particles. If the phase state would change, in particular under dry conditions, the particles should loose water and dry out. This would be observed in SMPS as number of particles would stay constant while the volume decreases. This would be observed during the preparation of the chamber experiment. In the procedure, the seed particles are injected. After injection the chamber is kept constant for 2-3 measurement cycles of the SMPS. After this period that takes 10-15 minutes the reaction is started. If the RH would dramatically effect the wall los this would be observed within the first 10-15 minutes while equilibrating the chamber. As this was not the case, this effect is regarded to be small for the used particles.

To address this issue a summary of the comment above is added to the manuscript. It now reads:

**Page 3, line 112:** *'For blank experiments all compounds were injected into the chamber, except the hydrocarbon. Notably, wall losses at RH = 50% was used as approximation also for 0% and 75% RH, although losses might change under those condition according to their phase state. According to previous studies wall loss might be small due to the short reaction time. According to McMurry and Grosjean, a 90 minutes reaction time would result in a 10% loss of particles, which is within the measurement uncertainty of the SMPS (McMurry and Grosjean, 1985). Additionally, seed particles were injected without a dryer. Consequently, they can be regarded as wet particles when they enter the chamber. Thus at RH =75% no additional loss is expected.*
*As the chamber is allowed to equilibrate for at least 10 minutes after seed injection, a dramatic wall loss under dry conditions would be directly observable in SMPS by a drastic decrease of particle volume with a constant particle number. As this was not observed it can be assumed that wall loss at RH = 0% is in the same manner as at 50 %.'*

+ L 106: add a comma between reaction and conditions to clarify the sentence meaning.

This sentence has been deleted during the present revision process.

+ L 117: The authors replied to the previous reviewer question that they use 1.0 g cm-3 as particle density because the real density of the particles is not known. The way this is phrased here suggests that 1.0 g m-3 is the true density (2 out of 3 readers misunderstood this information). It needs to be added in the text that in absence of any reliable density estimation, 1.0 g m-3 is used instead.

An additional information was added which reads

**Page 4, line 126:** *'In absence of reliable density estimation, an average density of 1 g cm$^{-3}$ was used to convert the SMPS measurement data into the increase in organic mass ($\Delta M$). This assumption was also made for OM and NPOM measurement. The assumed density was not changed with RH. As the ft between OM and $\Delta M$ stays almost constant, the RH seems not to affect the density in the conducted experiments.'*

+ L 118: was the water content accounted for when calculating the change in organic mass ☐M? I.e., was the whole volume as measured by the SMPS considered to be organic material? Or was a fraction subtracted according to the hygroscopic growth estimated for SOA at 50% and 75%?

To evaluate the contribution of water on the increase observed in SMPS, a comparison were made to OC/EC quantification from filter. Since both values fit very well, it can be assumed, that the particles grow upon partitioning and not strongly influenced by water uptake.

This has now been mentioned in the manuscript. The new text reads:

**Page 4, line 144:** '*The particle growth by water uptake was taken into account by collecting particles on filter and determine the content of organic material (OM). For most of the experiments it was found that both values ($\Delta M$ and OM) fit well indicating that particle growth is mainly caused by organics rather than water.'*

+ L 136: why is "biogenic and anthropogenic marker compounds" an italic subheading instead of adding it to the section heading?

This subheading was deleted.

+ L 130: IC-CD acronym not introduced

This is now introduced correctly.

**Page 5, line 157:** '*The amount of NO$_3^-$ was determined by ion chromatography coupled with conductivity detection (IC-CD) using an AS18 column combined with AG18 guard column.'*

+ L 144: HPLC acronym not introduced

This is now introduced correctly.

**Page 5, line 170:** *'A high-performance liquid chromatography (HPLC, Agilent, 1100 Series, Santa Clara, CA, USA) connected to a electrospray ionization time of flight mass spectrometer (microTOF, Bruker Daltonics, Bremen, Germany) was used for separation and quantification of marker compounds.'*

+ L 156: add the purity information for the pure chemicals from commercial sources.

The purity of chemical information is now added. This section now reads:

**Page 5, line 165:** *'The following chemicals were used as received: α-pinene, limonene and m-cresol (Sigma-Aldrich, St. Louis, USA, purity 99%, 97% and 99%), terebic acid (Sigma-Aldrich, St. Louis, USA, purity 99 %) and pinic acid (Sigma-Aldrich, St. Louis, USA, purity 99%).'*

+ How was the yield calculated. What assumptions were made?
A description how the values were calculatued are added to the manuscript. The text reads:

**Page 5, line 182:** '*The yield of the single compounds were calculated by taking the quantified amount from the filter, correcting for sampling volume. The numbers are given as fraction in formed organic mass.'*

+ L 177: Eq. (Eq. 1) correct typo
This typo was corrected.

+ L 178: it should be M0 both times. not M
+ L179: M0 is not listed
+ L 180: add "the" before mass yield

M0 is now corrected and the explanation is added. The text now reads:

**Page 6, line 215:**

$$Y = \sum Y_i = M_0 \sum \frac{\alpha K_{OM,i}}{1 + K_{OM,i} M_0} \qquad (Eq.\ 2)$$

*where*

*$\alpha$ is the mass yield of compound i*

*$K_{OM,i}$ is the partitioning coefficient of compound i*

*$M_0$ is the absorbing organic mass*

+ L 185: tiny tables in Fig 1. Even at 200% zoom, the numbers in the tables in Fig 1 are barely readable. Increase font size.
This might be an illustration problem by web page. While downloading the file and also checking the online version no problems with font sizes was there. Also the same font size was used for all Figures.

+ L 189: Can you really directly compare the parameters from a one product Odum fit with a two product fit? Have you tried fitting your data with a two product fit? How do the parameters compare?
We have obtained very good results with the one-product model with very high $R^2$ values. In the majority of cases with a $R^2$ of 0.99. This observation is in good agreement with Friedann and Farmer.

**Page 6, line 221**: '*By applying the one-product model approach, the fit produced very good results with $R^2 > 0.99$. The applicability of one-product models was also demonstrated by Friedmann and Farmer 2018. '*

+ L 200: It is not clear which of the 4 Fry et al. references is meant here.
It refers to Fry et al., 2014. The information is added. The text now reads:

**Page 6, line 217:** *'Although the values agree very well to the majority of the studies, it is still unclear why Fry and co-workers reported no SOA formation from α-pinene/NO₃ in the presence of seed particles (Fry et al., 2014).'*

+ L 212: "…proceeds via further reaction… " this is an ambiguous statement. Rephrase to clarify.
The sentence is re-phrased. It now reads:

**Page 7, line 251**: *'As it has been reported by Mutzel et al., 2016, the SOA formation of α-pinene /OH and limonene/OH is partly controlled via further reaction of myrtenal and limonaketone/endolim, respectively.'*

+ L 213: Isn't the limiting factor that the first-generation products are not of sufficiently low volatility to partition in the particle phase at the organic mass loading at that time? Further oxidation creates compounds of lower volatility which will contribute to particle mass. A different reaction pathway forms products with a different volatility distribution.
This was already discussed in the manuscript where we referred to earlier studies of exactly this point including a former study from our laboratory:

**Page 7, line 226:** *'The SOA formation from the OH-radical initiated reaction starts later than in the case of NO₃ for both systems, α-pinene and limonene. Such a long induction period is most likely caused by further reaction of first-generation oxidation products leading to SOA formation as it was demonstrated in previous studies (Ng et al., 2006, Mutzel et al., 2016). As it has been reported by Mutzel et al., 2016, the SOA formation of α-pinene /OH and limonene/OH could proceed via further reaction of myrtenal and limonaketone/endolim, respectively. The reaction of these first-generation oxidation products is the limiting factor for SOA formation and explains the delay in SOA growth (Mutzel et al., 2016). '*

+ L 216: In the presence of NOx, organonitrates should also form in the OH experiments.
Organonitrates might be formed but to a much lower extent. Then again, during OH radical reaction formed organonitrates might be photolysed. Therefore their contribution might be very low. We appreciate the reviewers comment but would prefer not to start a discussion on this topic to not de-focus.

L 219 yield from limonene + NO3
Space was added between limonene and NO₃.

+ L 220ff: 16-29% is close to 29-69%? I disagree. I also disagree with your statement that the NO3 reaction yields more SOA than ozonolysis.

It is clearly stated that this comparison refers to the lowest yields given for the ozonolysis in this text:

**Page 7, line 239:** *'Those values are close to the lowest values reported for limonene ozonolysis (Northcross and Jang, 2007, Chen and Hopke, 2010, Gong et al., 2018).'*

+ L 224 ff: You cannot compare the importance of NO3 vs OH vs O3 without stating how your experiment conditions relate to ambient conditions (what is the equivalent OH/NO3 exposure or oxidative age?). Unrealistically high NO3 values would distort your findings. AS you do not give these values one can only wonder.

The reviewer is correct that the concentration of OH and NO3 should be given. Therefore, the amount of OH and NO3 were calculated. These information were added into the experiment section (cf. earlier comment and reply). In general, it should be kept in mind, that the NO3 concentration present in the chamber (be 7.5 x $10^7$ molecules cm$^{-3)}$ are higher than under atmospheric conditions, but much lower than in other studies (Fry et al., 2014: 2.46 x $10^{10}$ molecules cm$^{-3}$).

Different from the reviewer we think a comparison can be done when the radical levels are given as they are now. For this reason the yield curve is given as a function of consumed hydrocarbon. This was done to avoid comparison of low vs high consumption experiments. Also this kind of illustration is used very often to compare experiments on an independent scale as it is normalised to the real consumption.

+ L 230ff: how much SOA mass was formed in your blank experiments (seed, oxidants but no VOC)? How does this "chamber background" SOA compare to the SOA mass formed in cresol+NO3?
As it is shown in Table 1, only 1 ug m$^{-3}$ was formed. As this is very low, it can be neglected.

+ L 241: "…was used." ? was used to form OH?
The photolysis of methylnitrite is a very common source for OH radicals.

+ L 249: "Aa discussed before…" N2O5 has not been mentioned anywhere before this point in this manuscript. Adjust the wording to reflect that.
The sentence was re-written. It now reads;

**Page 8, line 289:** *'As discussed in the section above, relative humidity has been suggested to influence SOA formation and yield for the $NO_3$-radical initiated reaction of VOCs.'*

+ L 253: "…- as to the authors' knowledge…" remove the "as"
Has been deleted. The text now reads:

**Page 8, line 220:** *'Only a very limited number of studies is available investigating the influence of RH on SOA formation originating from $VOCs+NO_3$ – the only ones, to the authors' knowledge, are as follows: Spittler et al., 2006, Fry et al., 2009 and Bonn and Moortgat 2002, Boyd et al., 2015'*

+ L 262: remove "Contrary". This implies a connection to the previous sentence that is not there.
'Contrary' has now been deleted. The text now reads:

**Page 8, line 280***: 'In the case of m-cresol/OH the SOA yield increases with humidity by a factor of 5.'*

+ L 274:"Fig 1"? should that not be Fig 2 here?
Growth curves are depicted in Fig. 2.

+ L 280: change "suggested" to "assumed" or "inferred"
It was changed accordingly.

The following two questions refer to the same topic/chapter and are discussed together:

+ L 281ff: Why would the behaviour of endolim create a RH dependent response? nothing in your explanation suggests a RH dependency.
+ L 291ff: same as for the previous paragraph. Why does this behaviour explain the observed RH dependence?

The reaction of VOCs such $\alpha$-pinene or cresol yields semi-volatile compounds that react further leading to the formation of products of sufficiently low vapour pressure. The contribution of first-generation oxidation products to the overall SOA formation has been described intensively in the past (Ng et al., 2006, Mutzel et al., 2016). Additionally, studies have demonstrated that the further reaction of first-generation oxidation products can lead to a strong increase with up to 20 – 40% in SOA formation after all precursor is consumed (Wang et al., 2018). Thus first-generation oxidation products are a key driver in SOA formation.

Therefore it cannot be ruled out, that their reaction is also affected by RH. As it was found that SOA marker compounds are formed from the reaction of first-generation oxidation products (Mutzel et al., 2016), the influence of RH and the product distribution of SOA marker compounds was already included into the text. The influence of RH on the partitioning of first-generation oxidation products is no included into the text. It now reads:

**Page 8, line 294:** *'The observed humidity dependencies could be caused by four main factors: (i) the uptake of the SOA marker compounds or their precursor compounds change as a function of the experimental conditions; (ii) the formation process of SOA marker compounds is directly affected by the experimental conditions; (iii) further reactions take place within the particle phase, and/or iv) the uptake behavior of the first-generation oxidation products might change with LWC. It remains a challenge to differentiate between all these factors because the observed dependency is most likely the result of a combination of all three factors. A discussion of factor i) to iii) is provided in the respective sections 3.4 Characterization of particle-phase chemical composition. The influence of RH on the uptake-behavior of first-generation oxidation products cannot be excluded. At it has been demonstrated in previous studies the uptake coefficient of first-generation oxidation products, in particular carbonyl compounds might depend on RH (Healy et al., 2009).'*

In the following section, a comprehensive discussion was given on the observed effect of RH on the reaction of limonene with $NO_3$ and *m*-cresol with OH, providing many possible explanations.

In the case of limonene/$NO_3$ is was found that the RH-dependency is most likely cause by an effect on the consumption. Therefore, an enhanced uptake of first-generation oxidation products is unlikely to be the reason because this woulnd´t explain the different consumptions. Thus an effect on or caused by endolim was excluded. During discussion in this section, different options were discussed, such as competition of limonene and endolim with $NO_3$. As this couldn´t be supported, an additional explanation was provided referring to a study by Boyd et al., 2015 that highlighted the importance of the competition between "$RO_2$+$NO_3$ dominant" and "$RO_2$+$HO_2$ dominant" reaction channels. During the discussion, it was hypothesized that limonene-originated $RO_2$ radicals are highly reactive and might represent an important sink for $NO_3$ which is in competition to limonene + $NO_3$.

In the case of m-cresol/OH it is stated in the manuscript that the effect of relative humidity on the partitioning of condensable products, such as methyl-nitro-catechol is most likely the reason. This suggestion is further supported in section 3.4 Characterization of particle-phase chemical composition because a significant increase of methyl-nitro-catechol was found at higher RH which is illustrated in Figure 7.

The link to the charaterisation of the particle-phase composition is now included into the manuscript. It now reads:

**Page 10, line 367:** *'This hypothesis is supported by the comprehensive characterisation of particle phase as discussed in section 3.4.'*

+ Regarding the RH dependence of Cresol+NO3 SOA yields. Have you considered a process analogue to the formation of isoprene SOA? There, smaller, semi-volatile but water-soluble compounds are taken up into the particle phase where they oligomerise and for low-volatility products. An aqueous phase enhance this by drawing the water-soluble compounds from the gas phase and also catalysing the condensed-phase reactions especially if SO42- ions are present (forming low volatility organosulfates).

Such a process as described by the reviewer seems to be very important for isoprene SOA. Within this study, such a process is unlikely to occur in an extented manner. The produced organic mass was pretty low, resulting in a SOA yield < 1%. Therefore, even if small functionalised compounds might be formed they seem to stay in the gas phase rather than partition into the particle phase. Therefore, this type of reaction is suggested to be less important for SOA originated from the reaction of cresol with $NO_3$.

+ L 310: Fig 7 is mentioned before figures 3-6.
The manuscript is changed and the text now reads:
**Page 10, line 353**: *'Thus, the delay might be caused by the effect of relative humidity on the partitioning of condensable products, such as methyl-nitro-catechol.*

+ Fig 7: MNC is -5-nitro in the caption but -6-nitro in the figure label and the main text.'

This was now corrected. The text reads now:

**Page 31, line 950:** *'Fraction of 3-methyl-6-nitrocatechol from the oxidation of m-cresol with OH. Please note, other compounds from the ASOA standard (Hoffmann et al., 2007) were not identified and due to the low SOA yield from $NO_3$-radical reaction, the concentration might be below the detection limit.'*

+ L 315: OC and WSOC are introduced here after they have already been used earlier. Ensure that acronyms/abbreviations are written out when they first occur and use the acronym/abbreviation consistently throughout the rest of the text.
This is not correct. Both abbreviations appear the first time on page 2 and they are explained.

+ L 319: What is the relationship between WSOC and NPOM. It reads here (and in the methods section) as if these two names are interchangeable. Is that the case? Is WSCO derived from the NPOM measurement?
WSOC is often expressed as NPOM, thus water-soluble organic carbon that cannot be purged during measurement. This is only caused by the method used for determination, cf. also the next comment and reply..

+ L 319: The authors speak of OC/EC and WSOC. But Fig 4 shows OM and NPOM. It is not explained how OM relates to OC/EC. It may seem trivial to the authors. But you cannot expect the general ACP reader to know all your measurement equipment and what quantities are measured/derived.

To avoid any misunderstanding only the terms OM and NPOM are now used in the manuscript.

+ L 320ff: When comparing the SMPS derived mass values with other directly mass based measurements, it is very important that the authors state their assumptions for the density, i.e., that it is set to 1.0 g cm-3 and is constant for all conditions. In comparable chamber experiments particle densities of 1.2 – 1.4 g cm-3 are typically derived. This means that the reported ☐M(SMPS) are most likely 20 – 40% too low. This has to be at least mentioned and considered when giving the uncertainties for the SMPS (caption of Fig 4). I recommend that the authors focus more on the qualitative trends ☐M(SMPS)

and OM (and NPOM, respectfully). This agreement may be an indication that the density does not change significantly with RH for these experiments.

As stated above, the density for transforming SMPS into produced organic mass a factor of 1 was used. This assumption was also made for OM and NPOM. A respective paragraph is added to the text. It reads:

**Page 4, line 126:** *'In absence of reliable density estimation, an average density of 1 g cm$^{-3}$ was used to convert the SMPS measurement data into the increase in organic mass ($\Delta M$). This assumption was also made for OM and NPOM measurement. The assumed density was not changed with RH. As the ft between OM and $\Delta M$ stays almost constant, the RH seems not to affect the density in the conducted experiments.'*

The measurement uncertainty for all measurement techniques applied in this study was already included in Figure caption 4. As this was already stated in the text, no additional information was included. Please see:

**Page 28, line 950:** *'Comparison of organic mass calculated from SMPS with an offline determined concentration of organic material (OM) and non-purgeable organic material (OM). Measurement uncertainties are given as follows 10% for SMPS measurements (Wiedensohler et al., 2012), 5% for OC/EC measurements (Spindler et al., 2004) and 10% for WSOM measurements (Timonen et al., 2010).'*

+ Fig 4: it is not directly clear what the % values refer to. Add this information to the Figure caption. The following sentence was added to the Figure Caption as suggested:

**Page 28, line 937:** *'The values of OM and NPOM are illustrated as fraction of the produced organic mass $\Delta M$ (expressed as % above the corresponding bars)'*

+ Fig 4 the uncertainty regarding the particle density needs to be mentioned when stating the uncertainty for the SMPS measurement. From the current wording it is not clear if the 10% already are supposed to take the density uncertainty into account.
It is stated in the Figure Caption that 10% are taken for measurement uncertainty caused by the SMPS. This is also described in Wiedensohler et al., 2012.

+ L 322: "..except mass originated from limonene" replace with "…except in the limonene experiments"
The sentence was changed accordingly to the comment. It now reads:

**Page 9, Line 340:** *'In general, the values agree, meaning the increase in organic mass corresponds to organic carbon and secondly, the majority of this mass is water-soluble, except in the limonene experiments.'*

+ L 324: replace "compound" with "precursor"
+ L 325: remove"is" (… with x% of org mass composed of…)

The sentence was changed according to the two comments. It now reads:

**Page 9, line 342:** *'In general, limonene with 22 – 36% of organic mass is the only precursor showing hints for water-insoluble material.'*

+ L 325ff: Have the authors considered that in the limonene+NO3 case a large fraction of hydrophobic products may be produced. An aqueous phase could then hinder the uptake of these compounds (quasi reverse co-condensation) which then stay in the gas phase leading to a lower SOA mass yield. It does seem odd that this happens in the NO3 case where the authors expect a higher organonitrate content. In general, the polar nitrate groups should make the molecules more hydrophilic

We thank the reviewer for this comment. This is a very interesting point. As depicted in Figure 4 limonene yields the highest fraction of water-insoluble OM. Thus, it cannot be excluded that hydrophobic compounds are formed and partition into the organic phase. Although this process needs to be considered, it is unlikely that this explains the current observation of the RH-dependency as this is related to a change in the consumption.

The formation of hydrophobic compounds as considered by the reviewer is now included into the main text

**Page 9, line 326:** *'As a contribution of ON is excluded the formation of that hydrophobic compounds that partition into the organic phase needs to be considered as potential explanation. As depicted in Figure 4 limonene yields the highest fraction of water-insoluble OM. Although this process needs to be considered, it is unlikely that this explains the current observation, Figure 2 clearly indicates a decreasing consumption when RH increases as a potential reason for lower SOA yields at higher RH.'*

+ L 326: Now the term "WSOM" is used. How does that relate to NPOM and WSOC? These inconsistencies make the manuscript hard to follow.

To avoid any misunderstanding, only the terms OM and WSOM are now used in the manuscript.

+ L 325ff: At least the name of the measurement principle for the peroxide measurement should be given here or better in the methods sections.
A sentence about the measurement principle is given. This reads:

**Page 10, line 357:** *'The method applies an iodometric detection by UV/Vis spectroscopy.'*

+ L 336: change "it" to "is"
The sentence deleted during revision process.

+ L 337: Is the "mostly dimer" assumption still valid considering the more recent information about the highly oxygenated material (HOM, (Ehn et al., 2014)) that is produced in auto-oxidation processes. Most of these compounds contain hydroperoxide groups but are not necessarily dimers.
In the work by Docherty et al., 2005 dimers and peroxyhemiacetals were suggested. Even that HOM contain hydroperoxide compounds it is unlikely that those compounds are determined by the applied iodometric peroxide test, as their lifetime might be much shorter than of dimers. As described in an earlier work by the authors (Mutzel et al., 2015) HOMs partition into the particle phase, but their contribution to the peroxide value might be low.

+ The absence of a RH dependence my be connected to the higher content of water-insoluble material in limonene SOA. If enough water insoluble material is present, a separate organic phase may form in which the peroxide compounds are "protected" from hydrolysis.
This could be a potential explanation. A corresponding sentence was added to the manuscript which reads:

**Page 10, line 398:** '*This indicates that organic peroxides are i) of a different nature than formed from α-pinene, ii) they originate from other reactions and/or iii) the high fraction of water-insoluble material a separate organic phase might be formed protecting peroxide from hydrolysis.*'

+ L 338: What is "the organic mass formed during the experiment" referring to (Morg)? The mass as derived from the SMPS measurement? Then the same thoughts regarding the uncertainty of the particle density apply here and should be clearly stated.

The 10% only refers to the measurement uncertainty of the SMPS and not the density assumption.

+ L 365ff: It is impossible to evaluate this comparison as no NOx values are given for the experiments in this study. Terms like low/medium/high are very subjective. (Isoprene SOA case).

We follow the terms generally used in the literature. For better clarity, initial NO concentrations are now included throughout the manuscript.

+ regarding possible artefacts in the peroxide measurements: How much H2O2 is left in the chamber? H2O2 uptake into particles may play an important role if very high concentrations of H2O2 were present (e.g. ppm level). The statement "blank filter were carefully checked" is not clear. Are these the filters collected after the blank experiments? Or are these blank filters? Will the uptake of H2O2 be the same on pure $(NH_4)_2SO_4/H_2SO_4$ particles as on mixed inoganic/organic?

The method to quantify the amount of consumed hydrocarbon by UV/Vis is a well-established. The applied technique was modified and comprehensively described in an earlier work of the group in Leipzig (Mutzel et al., 2013). Within this study, samples were taken on filter from $H_2O_2$ experiments. No hint for artefact formation was found. Secondly, the blank experiments conducted in the present study were comparable. The blank run was conducted by injecting seed, $H_2O_2$ and NO. After a reaction time of 90 minutes, filter samples were taken and analysed for their SOA-bound peroxide content. Also these experiments show no peroxide content. Thus for pure inorganic seed particles an effect of $H_2O_2$ originated from the injected $H_2O_2$ can be excluded.
Considering the last question if the uptake of $H_2O_2$ is the same for mixed organic/inorganic an answer is very difficult. If the organic content would control the $H_2O_2$ uptake, it could be expected that the detected amount of peroxides is the same for particles containing the same amount of OM. In the case of a-pinene/OH the content of OM is almost constant (around 5-6 $\mu g\ m^{-3}$) whereby the peroxide constant differs from 80 – 20%. A contribution of mixed organic/inorganic particles an $H_2O_2$ uptake cannot be excluded, but based on the present data set it can be regarded to be very small.

The added text reads:
**Page 10, line 379:** '*Although, earlier studies demonstrated that $H_2O_2$ injected into aerosol chamber does not cause artefacts, blank experiments were also conducted to exclude them. The blank run was conducted by injecting seed, $H_2O_2$ and NO. After a reaction time of 90 minutes, filter samples were taken and analysed for their SOA-bound peroxide content. Also these experiments show no peroxide content. Thus for pure inorganic seed particles an effect of $H_2O_2$ originated from the injected $H_2O_2$ can be excluded. Additionally, mixed organic/inorganic seed particles might be prone for partitioning of injected $H_2O_2$. If the organic content would control the $H_2O_2$ uptake, it could be expected that the detected amount of peroxides is the same for particles containing the same amount of OM. In the case of a-pinene/OH the content of OM is almost constant (around 5-6 mg $m^{-3}$) whereby the peroxide constant differs from 80 – 20%. A contribution of mixed organic/inorganic particles an $H_2O_2$ uptake cannot be excluded, but based on the present data set it can be regarded to be very small.*'

+ L 393ff: It is not clear to me what the marker compounds are normalised to. The SMPS derived organic mass? Or the OM? Or the HPLC Signal?
Additional information was added to the manuscript. The text now reads:

**Page 5, line 182:** '*The yield of the single compounds were calculated by taking the quantified amount from the filter, correcting for sampling volume. The numbers are given as fraction in formed organic mass.*'

+ L 406f: LWC and ALWC are not introduced. Is there a difference between the two quantities? Or is this another one of the inconsistencies that are spread throughout this manuscript?
Only ALWC is now introduced and used throughout.

+ L 422: HMWCs is not introduced.
Additional explanation is given.

**Page 13, line 468:** '*The last factor iii) to be considered involves further reactions of the SOA marker compounds to yield high molecular weight compounds (HMWCs) in the particle phase, as it has been often described in the literature (e.g. Gao et al., 2004, Tolocka et al., 2004, Müller et al., 2008, Yasmeen et al., 2010).*'

+ L 466: The O in ROS stands for oxygen – not organic! This term refers to the species creating oxidative stress in the system. These compounds can be organic peroxides. But it is definitely not referring to all organic. Non peroxy organic aerosol also have adverse health effects but via different mechanisms.
The wording was changed to "oxygen". It now reads:

**Page 14, line 513:** '*Such a dramatic increase might lead to an enhanced formation of reactive oxygen species (ROS) as determined here as peroxides and SOA.*'

+ L 482: The structure of the sentence "The formation of pinonic acid…" does not make sense in English. "proceed" is not the correct term when talking about chemical formation mechanisms. A phrase such as "Pinonic acid might be formed from the oxidation of pinonaldehyde that itself has been found for NO3-radical initiated reactions of a-pinene." is much clearer and describes the ongoing processes more precisely.

The sentence was changed according to the comment. It now reads:

**Page 14, line 307:** '*Pinonic acid might be formed from the oxidation of pinonaldehyde that itself has been found for $NO_3$-radical initiated reactions of α-pinene (Spittler et al., 2006).*'

+ L 491 "pointed out to be" do the authors mean "turned out to be"?

The sentence was deleted during the revision process.

+ L 495: The reactive species in the gas phase are only a reservoir for particulate matter if the reaction products have a sufficiently low vapour pressure or if other mechanisms form low volatility materiel (e.g. particvle phase oligomerisation as in the isoprene SOA case).

This is correct.